# The ultra-thin, minimally invasive surface electrode array NeuroWeb for probing neural activity

Jung Min Lee [1,5], Young-Woo Pyo [1,2,5], Yeon Jun Kim[2], Jin Hee Hong[2,3], Yonghyeon Jo [2,3], Wonshik Choi [2,3], Dingchang Lin [4] & Hong-Gyu Park [1]✉

Electrophysiological recording technologies can provide valuable insights into the functioning of the central and peripheral nervous systems. Surface electrode arrays made of soft materials or implantable multi-electrode arrays with high electrode density have been widely utilized as neural probes. However, neither of these probe types can simultaneously achieve minimal invasiveness and robust neural signal detection. Here, we present an ultra-thin, minimally invasive neural probe (the "NeuroWeb") consisting of hexagonal boron nitride and graphene, which leverages the strengths of both surface electrode array and implantable multi-electrode array. The NeuroWeb open lattice structure with a total thickness of 100 nm demonstrates high flexibility and strong adhesion, establishing a conformal and tight interface with the uneven mouse brain surface. In vivo electrophysiological recordings show that NeuroWeb detects stable single-unit activity of neurons with high signal-to-noise ratios. Furthermore, we investigate neural interactions between the somatosensory cortex and the cerebellum using transparent dual NeuroWebs and optical stimulation, and measure the times of neural signal transmission between the brain regions depending on the pathway. Therefore, NeuroWeb can be expected to pave the way for understanding complex brain networks with optical and electrophysiological mapping of the brain.

The development of various electrophysiological recording technologies[1–4] has yielded significant insights into the neurological underpinnings of behavior and cognition. For example, implantable probes, such as Si-based[5–7] or polymer-based multi-electrode arrays[8–12] with high electrode densities that can detect single-unit activities of individual neurons with high signal-to-noise ratios (SNRs), have played a major role in improving the understanding of neuronal networks; however, the invasive arrays of penetrating probes can damage brain tissue and produce unstable recordings due to long-term gliosis[13,14]. Even if the implantable probe is intended to be miniaturized with

tissue-like mechanical properties[8,9], severe lesions may occur during insertion into the target tissue. Because of these unavoidable issues of implantable multi-electrode arrays (iMEAs)[5,8–11], minimally invasive neural probes, such as surface electrode arrays (SEAs) or electrocorticography (ECoG)[15–19], have recently received increasing attention. Since SEAs are placed directly on the tissues or brain surfaces to interrogate the nervous systems without physical intervention, the related surgical approaches rarely result in an inflammatory reaction.

Flexibility and elasticity are the most important prerequisites for biocompatible SEAs that adapt to the complicated morphology and

[1]Department of Physics and Astronomy, and Institute of Applied Physics, Seoul National University, Seoul 08826, Republic of Korea. [2]Department of Physics, Korea University, Seoul 02841, Republic of Korea. [3]Center for Molecular Spectroscopy and Dynamics, Institute for Basic Science, Seoul 02841, Republic of Korea. [4]Department of Materials Science and Engineering, Johns Hopkins University, Baltimore, MD 21218, USA. [5]These authors contributed equally: Jung Min Lee, Young-Woo Pyo. ✉e-mail: hgpark@snu.ac.kr

convex surface of the brain[13]. Thus, a low Young's modulus and low bending stiffness, which play crucial roles in defining mechanical mismatch, are key considerations in the rational design of SEAs. In general, SEAs consist of three main components: (1) a recording electrode that detects neural signals, (2) an interconnector that transmits the detected signal to an external interface, and (3) a passivation layer that encapsulates the interconnector to ensure that it is electrically isolated. Conducting metals such as platinum (Pt) [8,18], iridium[20,21], and iridium oxide[22,23] are widely used for recording electrodes, due to their excellent ionic transfer to neural tissues. For interconnectors, Au films are commonly used due to their stability and fabrication feasibility[8,10,17]. In addition, to achieve a favorable tissue-device interface, soft materials such as polyimide[10,15,18], polydimethylsiloxane (PDMS) [24,25], SU-8[8,12], and hydrogels[26,27], have been developed for use in the passivation layer. These flexible SEAs are useful for finding the specific locations of cells and monitoring the function of the tissue less invasively.

Despite these advantages of SEAs, critical issues remain to be addressed. First, implementation of open lattice structures is difficult with soft materials such as hydrogel-based SEAs[26–28]. In particular, hydrogels have an electrically charged backbone and expand in water, requiring additional insulating layers[26–28]. Second, it is challenging to print PDMS accurately and fabricate the metal interconnects on it for SEAs[28]. Third, although interconnects of metal films are commonly used in soft material-based SEAs, they significantly increase the Young's modulus of the devices. In such cases, increasing the number of electrodes while maintaining device flexibility is a challenge. Last but not least, the overall thickness of the soft materials typically ranges from several to tens of micrometers, which makes them less suitable for conformal deployment on the curvilinear brain surface. The disadvantages of SEAs, which record weak neural signals with low spatial resolution[13], are further highlighted because such thick probes increase the distance between the neurons and probes[15].

In this work, to address these issues, we developed an ultra-thin SEA neural probe (the "NeuroWeb") consisting of hexagonal boron nitride (h-BN) and graphene (Gr) that leveraged the strengths of single-unit spikes detection of iMEAs and minimal invasiveness of SEAs. An open lattice structure with a total thickness of ~100 nm allowed the detection of high-quality neural signals on the surface of the live mouse brain, forming a conformal and tight interface. The improved mechanical properties of this minimally invasive probe were assessed through flexibility and adhesion experiments. We also showed the possibility of vertical stacking of high-density electrodes for multiplexing. In addition, in comparison with 1 μm-thick polymer-metal SEA, our probe showed significant increases in spike amplitude and number of detected neurons as well as excellent visible-light transmittance. Furthermore, the optogenetics experiment using dual NeuroWebs enabled simultaneous recordings of neural signals from two distinct regions of the central nervous system (CNS), the somatosensory cortex and the cerebellum. We believe that NeuroWeb represents a unique type of neural probe capable of optical and electrophysiological mapping of the brain surface with high spatial and temporal resolution.

## Results

### Design and fabrication of NeuroWeb

NeuroWeb is a minimally invasive neural probe that detects action potentials on the surface of the live mouse brain (Fig. 1a). NeuroWeb consists of the open lattice structures of the active region, supporting region, and metal interconnects, serving different functions. First, the active region contains 32-channel Pt electrodes (Pt/Au joint/Pt), which are connected to Gr lines sandwiched with top and bottom h-BN insulating layers (Fig. 1b). The Pt electrodes are exposed from the open lattice structure for neuron detection. The top SU-8 layer increases the mechanical strength of the active region. This active region consisting of SU-8/h-BN/Gr/h-BN is designed to have a thickness of 100 nm.

Second, the supporting region consisting of the SU-8/Au/SU-8 open lattice structure allows the active region to spread out without tangling. The active region is attached to the brain surface without wrinkles by leveraging the strength of the supporting region. Third, the metal interconnects of Au lines electrically connect the Gr lines and the input/output (I/O) pads. I/O pads are eventually connected to an external recording instrument. The supporting region and metal interconnects have a thickness of ~1 μm.

The thicknesses of Gr and h-BN are in the order of a few and tens of nm, respectively, which depend on the number of layers; thus, an ultra-thin NeuroWeb can be implemented, demonstrating the following possibilities. First, as the bending strain decreases linearly and the inelastic materials become flexible with decreasing thickness[29], NeuroWeb adheres firmly even to corrugated surfaces. Second, the improved flexibility and adhesion of NeuroWeb increase the contact area between electrodes and brain tissue, enabling more efficient detection of neural signals. Third, due to the transparency of NeuroWeb, fluorescent images of neural activities can be optically monitored and captured in real-time without obstructing the field of view. Fourth, well-designed optogenetics experiments can be carried out, monitoring intricate neuronal pathways across multiple brain areas.

As a first step, to assess the feasibility of utilizing multilayer h-BN as a passivation layer for rational probe design, we measured the I–V characteristics of Gr lines covered with multilayer h-BN of ~30 nm thickness in single-channel or cross-channel configurations with and without a phosphate buffered saline (PBS) solution on the h-BN (Fig. 1c). Only a negligible increase in current leakage was observed, from ~390 to ~500 nA at 1 V, when a PBS solution was put on top of the h-BN layer (red dots in Fig. 1c). No frequency dependence was also measured. This measurement result shows that the h-BN can serve as an effective passivation layer in neural probes.

NeuroWeb was then fabricated by a series of conventional photolithography steps. We show four main steps in the fabrication of the active region (Fig. 1d and Supplementary Fig. 1). First, a bottom Pt electrode with a diameter of 20 μm and thickness of 50 nm was defined on the Ni sacrificial layer. The multilayer h-BN film with an average thickness of ~30 nm was transferred and patterned on the same substrate to make a bottom insulating layer (Fig. 1d, i). Second, Au joints were fabricated on Pt electrodes for a solid mechanical connection between the electrodes and h-BN ribbons (Fig. 1d, ii). Third, three layers of monolayer Gr were transferred and patterned to electrically connect Pt recording electrodes to the Au lines in the metal interconnects (Fig. 1d, iii). Last, the top Pt electrodes were fabricated in the same manner as the first step to form double-sided electrodes. A SU-8/ top h-BN layer covered Gr interconnectors to electrically isolate them from the external noise (Fig. 1d, iv). We note that it makes no difference which side of NeuroWeb is placed on the brain surface because we employ double-sided electrodes that can detect neural activity regardless of electrode orientation[30]. A more detailed fabrication procedure for the supporting region and metal interconnects was introduced in Supplementary Fig. 1.

Figure 1e and f show a fabricated NeuroWeb and its structural details, respectively. We examined the structural, mechanical, and functional properties of the device. First, a cross-sectional transmission electron microscope (TEM) image shows the structural components of NeuroWeb (Supplementary Fig. 2). The total thickness of NeuroWeb was corroborated by an atomic force microscope (AFM) imaging, which revealed an average thickness of ~100 nm (Fig. 1g). Second, bending experiments exhibited the high mechanical reliability and stability of the device (Supplementary Fig. 3). Third, to evaluate the functionality of NeuroWeb as a neural probe, the impedance values of the recording electrodes were measured in 1× PBS at 1 kHz (Supplementary Fig. 4a) and 20 Hz to 4 kHz (Supplementary Fig. 4b). The average impedance of ~540 kΩ is low enough to sensitively detect neural signals[13]. This experimental result, in conjunction with the I–V

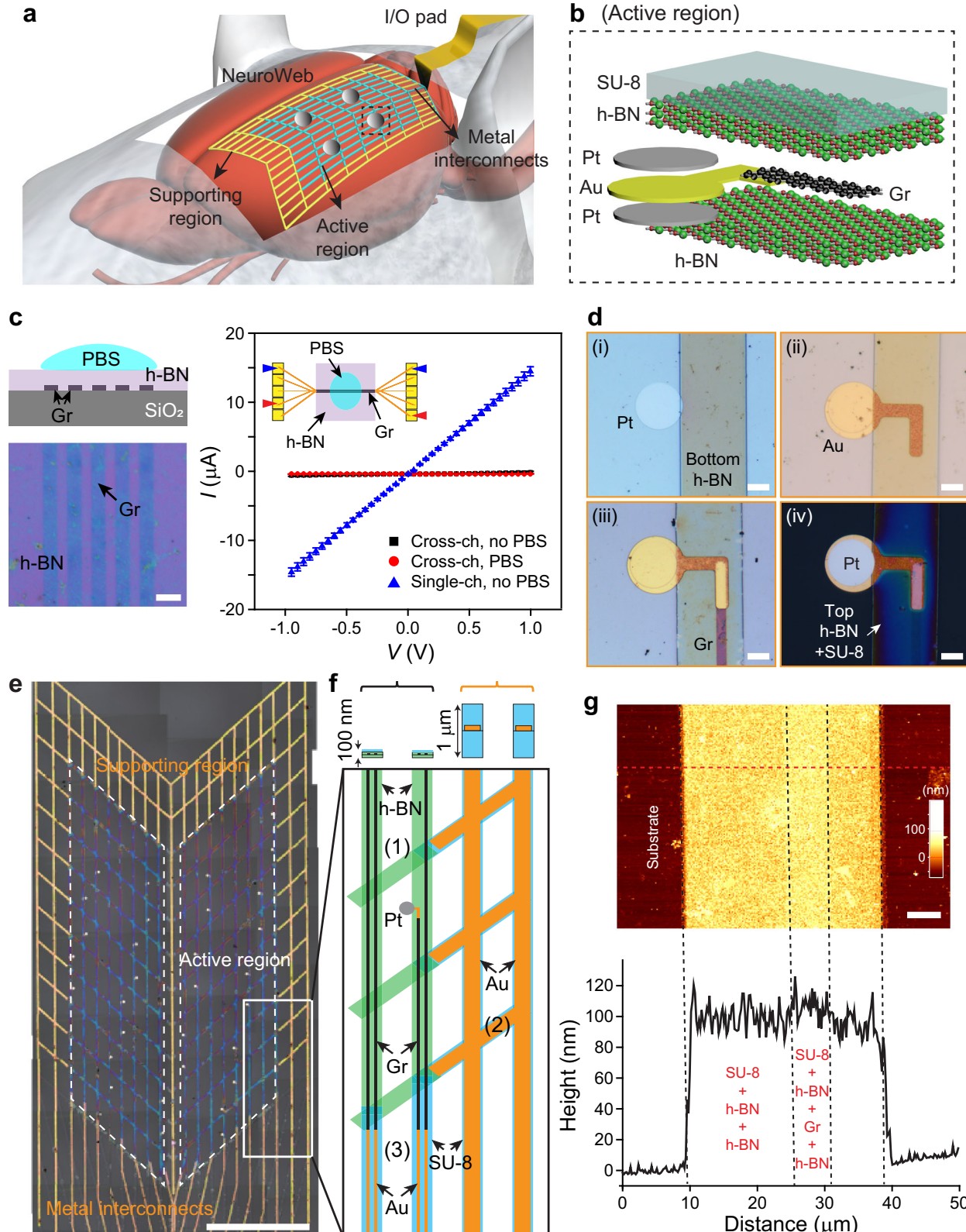

characteristics shown in Fig. 1c, demonstrates direct evidence of h-BN passivation. We note that NeuroWeb can record high SNR signals while being less affected by signal attenuation due to shank capacitance[31,32] (Supplementary Fig. 5).

### Flexibility, adhesive strength, and stacking of NeuroWeb

The mechanical properties of NeuroWeb, flexibility and adhesion, were systematically investigated to assess whether this probe can be conformally and closely attached to uneven surface morphologies. First, for the flexibility experiment, we fabricated the periodic SU-8 line patterns with a width of 10 µm and height of 5 µm on the Au film-coated substrate at various spacing distances ($d$) from 50 to 80 µm, and then transferred NeuroWeb onto the substrate (Fig. 2a). For comparison, a standard polymer-metal SEA (SU8-Au SEA) was also transferred to the same substrate (Fig. 2b). The SU8-Au SEA we fabricated has a thickness of ~1 µm (Methods), which is relatively thinner

**Fig. 1 | Overall design of NeuroWeb. a** Schematic illustration of NeuroWeb on the mouse brain surface. NeuroWeb consists of the active region, supporting region, and metal interconnects. The detected action potentials using NeuroWeb are sent to an external interface via I/O pads. **b** Close-up view of the black dashed box in (**a**), which consists of the Pt recording electrode (gray), Au joint (yellow), Gr line (black), top and bottom h-BN insulating layers (green), and SU-8 (light blue). **c** Left, schematic (top) and optical image (bottom) of Gr lines (5 μm × 3.1 mm) on a SiO₂/Si substrate, covered with a multilayer h-BN. Scale bar, 5 μm. Right, measured I–V characteristics of Gr/h-BN heterostructures in single-channel (blue dots; $N = 3$) and cross-channel with and without a PBS solution on h-BN (red and black dots; $N = 3$, each). The inset shows the schematic for the I–V measurement. Error bars denote s.d. **d** Optical microscope images of key fabrication steps in the active region: (i) patterning of the bottom Pt electrodes and bottom h-BN; (ii) patterning of Au

joints; (iii) patterning of Gr lines; (iv) pattering of the top Pt electrodes and SU-8/top h-BN. Scale bars, 10 μm. **e** Optical microscope image of a fabricated NeuroWeb. The active region (white dashed parallelograms) is surrounded by the supporting region and connected to the metal interconnects. Because the 2D materials used in NeuroWeb are ultra-thin, their colors can be observed due to interference with the substrate[61]. Scale bar, 1 mm. **f** Schematic of the white box in (**e**), which shows the (1) active region, (2) supporting region, and (3) metal interconnects, including h-BN (green), Gr (black), Pt (gray), SU-8 (light blue), and Au (orange). The top inset shows the side-view schematic. The black and orange brackets indicate the active and supporting regions, respectively. **g** AFM image measured in a SU-8/h-BN/Gr/h-BN ribbon. Bottom, height profile across the red dashed line of the image. Scale bar, 5 μm. Source data are provided as a Source Data file.

than other existing probes and commercial probes. Except for the thickness ( ~ 100 nm vs. -1 μm), the other dimensions of NeuroWeb and SU8-Au SEA are the same.

As shown in the SEM images (Fig. 2c and Supplementary Fig. 6), NeuroWeb was bent along the SU-8 structures with $d$ ranging from 50 to 80 μm, and the recording electrode was securely attached to the Au-coated substrate. However, the electrodes of SU8-Au SEA did not touch the substrate due to the relatively thick SU-8 layer and the stiff Au interconnects embedded in SU-8 (Fig. 2d and Supplementary Fig. 7). These results can be quantitatively analyzed by measuring the resistance between the electrode and the Au-coated substrate (Fig. 2e and Supplementary Fig. 8). The resistance decreased from 195.4 to 47.4 kΩ with increasing $d$ in the case of NeuroWeb. Due to the remarkable flexibility of NeuroWeb, the recording electrode showed efficient electrical contact with the Au-coated substrate containing the obstruction, and the low resistance was measured. In contrast, the SU8-Au SEA showed infinite resistance.

Next, we experimentally examined the adhesion strengths of ~100 nm-thick NeuroWeb and -1 μm-thick SU8-Au SEA. To mimic the in vivo situation of a probe transferred to the brain surface, NeuroWeb and SU8-Au SEA with similar contact areas were attached to the slide glass in a Petri dish, and deionized (DI) water was applied to induce stress at the interface between the probes and the slide glass. We measured the duration for which they were adhered to the slide glass in DI water. A series of photographs clearly showed that the active region of NeuroWeb was still intact and was not detached from the glass surface even after one day (top panel, Fig. 2f). On the other hand, SU8-Au SEA was completely released from the glass in an average of 28.9 s after pouring DI water (bottom panel, Fig. 2f). These results indicate that NeuroWeb has superior adhesion strength to the glass surface, in comparison with a standard polymer-metal SEA.

To better understand the origin of the adhesion strength rather than comparing the experimental results, we performed numerical simulations of adhesive stress by detaching a probe from the substrate. To simplify the simulation, h-BN films with different thicknesses ranging from 60 to 1000 nm were examined while their exact forms on the substrate were identified using SEM images in Supplementary Fig. 6b (Supplementary Fig. 9). The patterned substrate shown in Fig. 2a was employed, but $d$ was set to 60 μm. The simulation showed that adhesive stress decreased rapidly with increasing h-BN thickness (Fig. 2g), consistent with the finding that the ultra-thin NeuroWeb with a thickness of ~100 nm exhibits a considerable increase in adhesive stress at the interface.

Another strength of ultra-thin NeuroWeb is that the number of electrodes can be easily expanded by stacking additional layers of h-BN and Gr in the same location. Initially, the 1st-layer Gr interconnectors were sandwiched by the bottom and top h-BN layers. For the stacking strategy, the 2nd-layer Gr interconnectors patterned on the top h-BN layer were covered with another top h-BN (Fig. 2h). We designed the stacked NeuroWeb with no Au joints to avoid potential leakage between the Au joints and the 2nd-Gr layer. The total thickness of the

added layers (h-BN + Gr) in the fabricated stacked NeuroWeb was only ~40 nm, which made it difficult to distinguish between the 1st and 2nd-Gr layers from a top perspective because the 2nd-Gr layer was placed perfectly on the 1st-Gr layer (Fig. 2i). In addition, the stacked Neuro-Web has comparable impedance values to the non-stacked original NeuroWeb (Supplementary Fig. 4), indicating that the stacked Neu-roWeb can still be used as a brain surface probe without electrical leakage. Furthermore, the flexibility of the stacked NeuroWeb was also examined in the same experiments shown in Fig. 2a. Despite the slight increase in thickness, the stacked NeuroWeb made effective electrical contact with the Au-coated substrate (Supplementary Fig. 10). The resistance of the stacked NeuroWeb was measured to be similar to that of the original NeuroWeb at $d = 80$ μm (Fig. 2e). Taken together, the multiplex scalability of NeuroWeb is straightforward, allowing for an increase in electrode density within the same dimensional constraints by stacking layers of Gr and h-BN.

## Electrophysiological recordings with NeuroWeb on the live mouse brain surface

Next, to detect action potentials on the mouse brain surface, Neuro-Web was placed on the cortical surface after removing the skull ( ~ 3 × 3 mm²) and dura mater. The I/O portion of NeuroWeb was electrically contacted to a Flexible Flat Cable (FFC) via a direct-contact method[33] for signal transmission from 32 Pt recording electrodes to an external interface (Fig. 3a). The magnified optical microscopy image showed wrinkle-free adhesion of the whole active region to the mouse brain surface with a diameter of -2 mm (Fig. 3b). The exceptional transparency of the BN-Gr ribbons allowed for clear observation of the active region. Only Pt recording electrodes were visible on the surface of the brain (white arrows, Fig. 3c).

For long-term in vivo recording, NeuroWeb was covered with 3 mm-diameter round glass and the interface was sealed with dental cement. The light weight of the entire interface ( ~ 1.5 g), including the NeuroWeb, head stage, and FFC, and the compact footprint of the NeuroWeb and head stage allow the mouse to move freely (Supple-mentary Fig. 11). Electrophysiological recordings from the surface of primary somatosensory cortex (S1) at day 1 after surgery highlight the following key features. First, we performed local field potential (LFP) analyses (Fig. 3d–f), to evaluate neural population activity in the S1 using NeuroWeb. The original neural signals (Supplementary Fig. 12) were band-pass filtered from 0.3 to 300 Hz to isolate the LFPs. A representative set of four channels (Ch 6–9) shows LFPs with mod-ulation amplitudes of -250 μV (Fig. 3d). To investigate the change in frequency content over time, we obtained a spectrogram that illus-trates the time-frequency distribution of LFP power in Ch 8 (Fig. 3e): oscillations in the low frequency range (1–8 Hz) were observed in the S1. In addition, the LFP power spectrum at each channel exhibited oscillations in the delta frequency range (0.5–4 Hz) (Fig. 3f), which originated from neocortical and thalamic neural circuits[34].

Next, band-pass filtered (250–6000 Hz) data from NeuroWeb exhibits the capability to detect single-unit neural activities with the

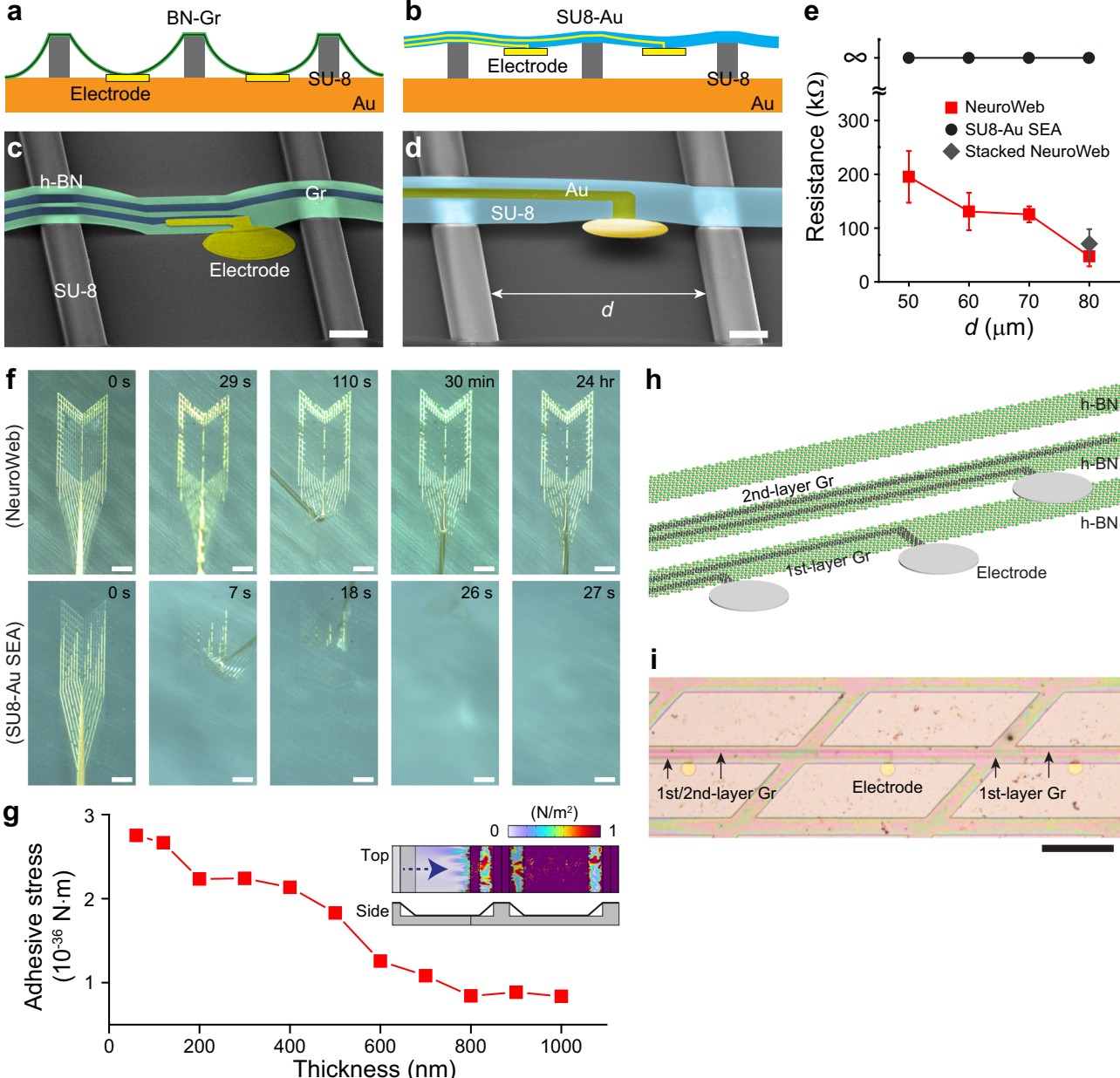

**Fig. 2 | Mechanical properties and stacking of NeuroWeb. a** and **b** Schematics of the flexibility experiments for NeuroWeb (**a**) and standard polymer-metal (SU8-Au) SEA (**b**). Both probes were transferred to periodic SU-8 lines with varying distances (50, 60, 70, and 80 μm; $N = 3$, each) on the Au-coated substrate. **c** and **d** SEM images of NeuroWeb (**c**) and SU8-Au SEA (**d**) on the Au-coated substrate with SU-8 lines with a distance (*d*) of 60 μm. Scale bars, 10 μm. **e** Measured resistance as a function of *d* (50–80 μm) for NeuroWeb (red square) and SU8-Au SEA (black circle). The measured resistance of stacked NeuroWeb in (**i**) was also displayed at *d* = 80 μm (gray diamond). Error bars denote s.e.m. The resistance measurement was repeated $N = 3$ in each case. **f** Series of photographs showing the adhesive strengths of NeuroWeb for 24 h (top) and SU8-Au SEA for 27 s (bottom). DI water was poured until the probes on the slide glass were fully immersed. Scale bars, 1 mm. Adhesive test was repeated with $N = 3$ independent samples. **g** Calculated adhesive stress as a function of the thickness of the transferred h-BN, when a periodic pattern was introduced with a spacing of 60 μm on the substrate. The insets show the calculated top-view and side-view von Mises stresses at a h-BN thickness of 60 nm. **h** Schematic showing the stacking strategy in NeuroWeb. By stacking additional layers of h-BN (green) and Gr (black) to the original NeuroWeb, the density of electrodes is doubled. **i** Optical microscope image of the stacked NeuroWeb according to the schematic in (**h**). Scale bar, 100 μm. Source data are provided as a Source Data file.

highest amplitude of ~122 μV and SNR of ~31 at day 1 (Fig. 3g and Supplementary Fig. 13a). Spike sorting analysis was performed to identify the number of recorded single units from each channel (Fig. 3h). Most of the recording electrode channels showed three or four single-unit spikes, as indicated by the different colors of the spike waveforms. The shape and amplitude of the spikes are influenced by the position and orientation of the electrode surface relative to neurons[2,8,35]. In addition, the average amplitudes from 30-channel electrodes of NeuroWeb at day 1 and 7 were ~38 μV and ~53 μV,

respectively (Fig. 3i and Supplementary Fig. 13b). The average SNRs were ~9.3 and ~10.7, respectively (Fig. 3j and Supplementary Fig. 13c). This difference in the average SNRs between day 1 and 7 was not statistically significant ($P > 0.05$, paired-sample *t*-test; Supplementary Fig. 13d). Furthermore, the three neurons recorded from Ch 8 were tracked from day 1 to 7 to show the stability of the waveforms (Fig. 3k). Principal component analysis (PCA) also proved that the waveforms of the recorded units were well preserved during prolonged recording (Supplementary Fig. 13e).

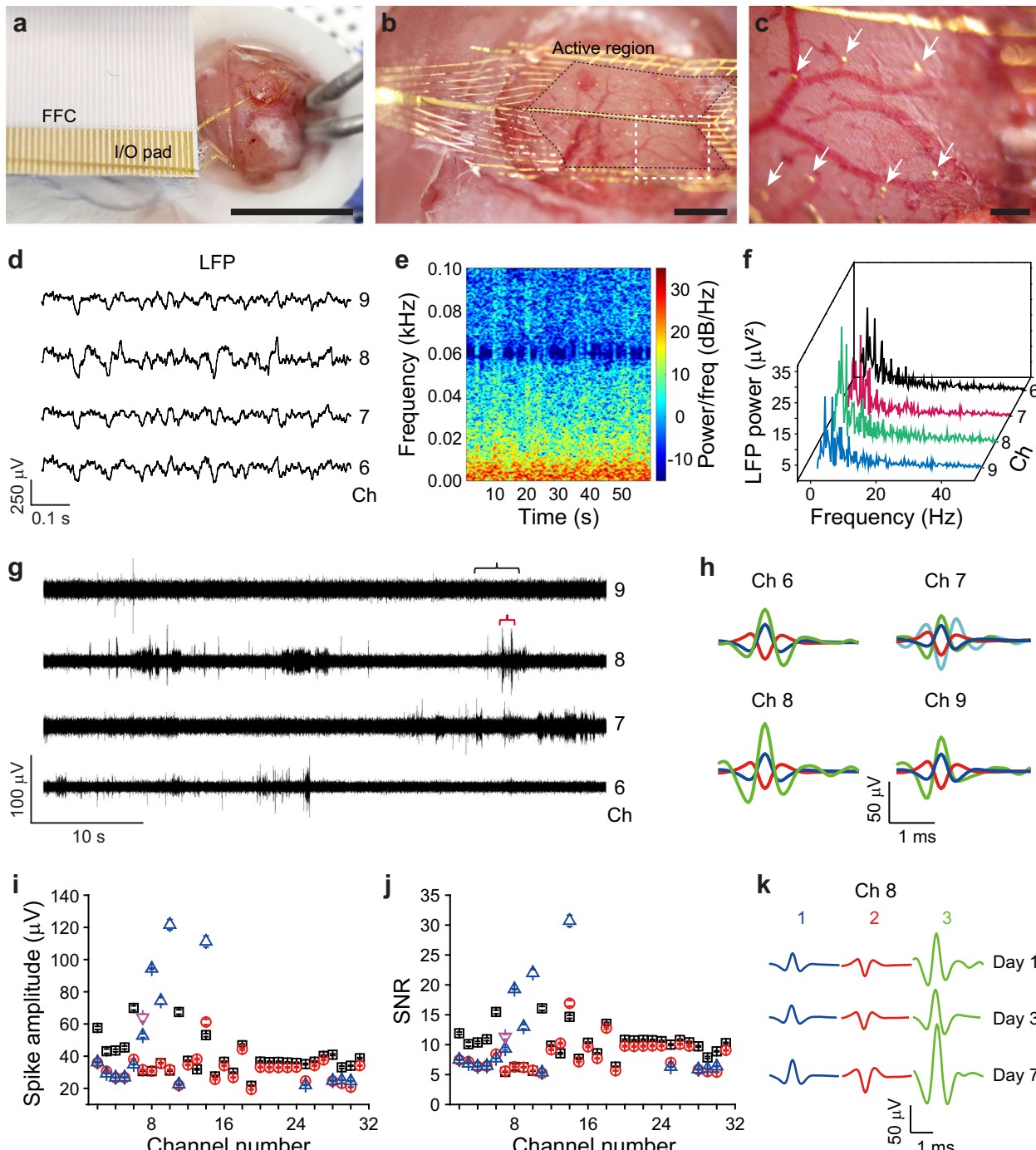

**Fig. 3 | In vivo experiments using NeuroWeb. a** Optical image of NeuroWeb attached to the brain surface of a live mouse. The 32 Pt electrodes of NeuroWeb are electrically connected to FFC via I/O pads. Scale bar, 1 cm. **b** Magnified image of the NeuroWeb active region (black dashed lines), which is well-attached to the brain surface. Scale bar, 1 mm. **c** Further magnified image of the white dashed box in (**b**). Seven electrodes (white arrows) are visible, whereas the transparent BN-Gr ribbons are not visible to the naked eye. Scale bar, 100 μm. **d** LFPs in four channels (Ch 6–9) from the S1 at day 1 after surgery, which were band-pass filtered (0.3 – 300 Hz) from the original neural signals. **e** Spectrogram (Ch 8) showing the time-frequency distribution of LFP power in the S1. **f** LFP power spectra (Ch 6–9) as a function of frequency. **g** Four band-pass filtered (250–6000 Hz) extracellular spike recordings from the S1 at day 1 after surgery. The whole trace data are shown in Supplementary Fig. 13. The black and red brackets indicate the regions of the LFP trace in (**d**) and the original trace in Supplementary Fig. 12, respectively. **h** Spike sorting analysis of the recordings shown in (**g**). **i** and **j** Spike amplitudes (**i**) and SNRs (**j**) of the recorded single-unit action potentials for each channel at day 1 after surgery. Each distinct color corresponds to a neuron uniquely identified for each channel. Error bars denote s.e.m. **k** Time-course analysis of the three spike waveforms detected from Ch 8 in (**h**). Source data are provided as a Source Data file.

We note that it was possible to record single-unit signals from the surface of the cortex in previous studies, but the recording yield was very low. NeuroWeb reliably detects high-yield single-unit spikes because it is >20 times thinner than the probes used in earlier works[15,17,36]. Consequently, the recorded average amplitudes and SNRs

were much higher in our experiment than the values in conventional SEAs, and similar to those in invasive implantable probes (Supplementary Table 1)[8,37]. Furthermore, stable electrophysiological recordings were investigated for a week, based on statistical and time-course analysis of average spike waveforms. Therefore, NeuroWeb shows

superior performance comparable to implantable probes, while causing minimal brain tissue damage.

In addition, to verify the functionality of NeuroWeb for detecting neurons from the brain surface, we performed whisker stimulation, a well-established and behaviorally effective technique[38,39]. Touching the whisker sends signals to the contralateral brain via the brainstem, thalamic neurons, and trigeminal ganglion. We placed NeuroWeb on the left S1 (barrel cortex) and measured neural signals, while repeatedly and periodically touching the whiskers of an awake mouse (Supplementary Fig. 14a). The measured extracellular action potential traces showed the single-unit spike activities evoked by whisker stimulation at different intervals of 4.04, 6.04, 8.63, and 10.07 s (Supplementary Fig. 14b). In the spike raster plot of 20 stimulation trials, the spike events showed increased firing rates following whisker stimulation (Supplementary Fig. 14c). A clear correlation between whisker stimulation and evoked spikes was observed, suggesting that single-unit activity can be assigned to biological activity that responds to stimulus.

## Optical properties and electrophysiological analysis of NeuroWeb

Next, to more quantitatively evaluate the unique properties of NeuroWeb, such as high transparency of the active region and robust detection of neural signals, we compared them with those of a standard polymer-metal SEA (SU8-Au SEA) with a thickness of ~1 μm. To this end, NeuroWeb or SU8-Au SEA was placed onto the live mouse brain surface (Fig. 4a). The optical images of the two probes exhibited very different features from each other (Fig. 4b). The brain surface was clearly visible underneath the transparent NeuroWeb but not underneath the SU8-Au SEA with dense Au lines. Moreover, wrinkles were observed in the SU8-Au SEA; the mechanical mismatch between the brain and the SU8-Au SEA, which has relatively thick SU-8 layers and stiff Au interconnectors, resulted in these wrinkles during the transferring and drying processes.

To investigate the transparency of NeuroWeb and SU8-Au SEA systematically, we conducted in vivo imaging of myelinated axons[40] through these probes on the brain surfaces of live mice (Supplementary Fig. 15). After both probes were transferred to the exposed cortex of an 8-week-old Thy1-EGFP line M mouse, a 100 μm-thick cover glass was glued to the unremoved surface of the mouse skull using dental cement to open an observation window. In the measured confocal fluorescence images (Fig. 4c), the BN-Gr ribbon (SU8-Au ribbon) covered 21.4% (23.7%) of the field of view at a depth of 16 μm (26 μm). In particular, the region underneath the BN-Gr ribbon merged in myelinated axons without noticeable disruption, making it difficult to locate the BN-Gr ribbon. On the other hand, the region underneath the SU8-Au ribbon was much darker than the surroundings, and the myelinated axons in that region were less visible than in the case of NeuroWeb.

The fluorescence intensities of myelinated axons underneath the BN-Gr and SU8-Au ribbons were quantitatively compared with each other. Normalization of intensities was performed using the values just outside each ribbon (Supplementary Figs. 16 and 17). The measured intensities were ~1 and ~0.48 for NeuroWeb and SU8-Au SEA, respectively (Fig. 4d). This indicates that while the BN-Gr ribbon was transparent, the fluorescence intensity with the SU8-Au ribbon was almost halved. Furthermore, the transmittances of NeuroWeb and SU8-Au SEA were measured to be 97.2% and 86.2%, respectively, when the ribbons were placed in the middle of a circular-shaped field of view with a diameter of 100 μm (inset, Fig. 4d). We also measured a uniform and high transmittance of NeuroWeb over a wide visible wavelength range (480–780 nm), with an average transmittance of 96% (Supplementary Fig. 18). These findings support the utility of the NeuroWeb for optical recording and stimulation, demonstrating the potential for simultaneous optical and electrophysiological mapping of complex neural activity.

In addition, quantitative comparisons of electrophysiological recordings using NeuroWeb and SU8-Au SEA revealed several key features as follows (Supplementary Figs. 19 and 20). First, the average spike amplitudes were ~33 and ~15 μV for NeuroWeb (N = 90 channels) and SU8-Au SEA (N = 73 channels), respectively, on day 3 after surgery (Fig. 4e). The ~2.2-fold higher spike amplitude in NeuroWeb than in SU8-Au SEA was statistically significant ($P < 0.0001$). Second, by comparing the number of spike-sorted single neurons per channel in NeuroWeb and SU8-Au SEA, we found that NeuroWeb detected on average ~1.3 times more single units per channel than SU8-Au SEA (Fig. 4f and Supplementary Figs. 21 and 22). This difference was also statistically significant ($P < 0.0001$). Third, the performance of NeuroWeb (N = 90 channels) and SU8-Au SEA (N = 73 channels) was examined for a week (Fig. 4g). The average spike amplitude of NeuroWeb remained more than twice that of SU8-Au SEA after 3 days. Taken together, because of the superior contact with the brain surface, NeuroWeb steadily detected stronger neural signals than SU8-Au SEA.

## Investigation of neural circuits in the brain using dual NeuroWebs with optical stimulation

Finally, we ask whether NeuroWeb can detect neural activity in more wrinkled areas of the brain and resolve complex neural circuits across brain regions. As a proof of concept, we investigate the neural connections between the somatosensory cortex (S1) (ML: 3.4, AP: −2.3, DV: 1.1) and the cerebellum (Cb) (ML: 0, AP: −6.3, DV: −0.9)[41], using dual NeuroWebs with optical stimulation. NeuroWebs A and B are attached to the S1 and Cb of a transgenic mouse with the expression of ChR2 and yellow fluorescent protein (Thy1-ChR2-YFP), respectively, to record neural signals evoked by 488 nm laser stimulation of the S1 or Cb (Fig. 5a). During the surgery, the skull and dura mater were carefully removed from S1 and Cb in order to attach two NeuroWebs (Fig. 5b). We notice that the Cb has many ridges or folds that give it a wrinkled appearance. Each NeuroWeb was electrically connected to the corresponding 32-pin FFC (FFC-A or FFC-B) using a direct-contact method[33], and a fiber-coupled laser diode was used to selectively stimulate the S1 and Cb (Fig. 5c). In this experiment, we performed multi-cell stimulation to study optically evoked neural activity and circuitry in different CNS regions of the S1 and the Cb (Methods).

Both NeuroWebs firmly adhered to the brain surface, particularly the wrinkled surface of the Cb, and were able to detect neural signals in multiple channels (Fig. 5d). The electrophysiological recordings from NeuroWebs A on the S1 and B on the Cb with optical simulation reveal the following key features. First, mice were under anesthesia, inducing suppressed neural activity[42]. The channels of NeuroWebs A and B showed almost no spontaneous neural activity. Second, an optically evoked signal was observed at laser power densities >1.30 mW/mm$^2$, while no signal was observed at lower levels. We varied laser power densities up to 1.59 mW/mm$^2$ to stimulate only neurons near the electrodes and minimize thermal effects[43]. Third, optically evoked signals were detected in response to the laser cycle. For example, when the Cb was optically stimulated, the signals recorded in NeuroWeb B were only observed in the on states of the laser pulses (red traces, Fig. 5d). Stronger signals were detected at the electrodes near the neurons being stimulated, while there is no time difference between the evoked signals. Fourth, as the laser power density increased from 1.30 to 1.59 mW/mm$^2$, the amplitudes of the optically evoked signals are not significantly different (Fig. 5e). The absence of a change in the amplitude of the detected signal with increasing laser power in such a low laser power regime suggests that the signal is not caused by photoelectric artifacts[44,45]. Fifth, neural signals were detected even in a non-stimulated brain region (Fig. 5d and Supplementary Fig. 24). When the Cb was optically stimulated, signals with the same period as laser pulses were detected in NeuroWeb A on the S1 with a latency from the optical stimulation (black traces, Fig. 5d).

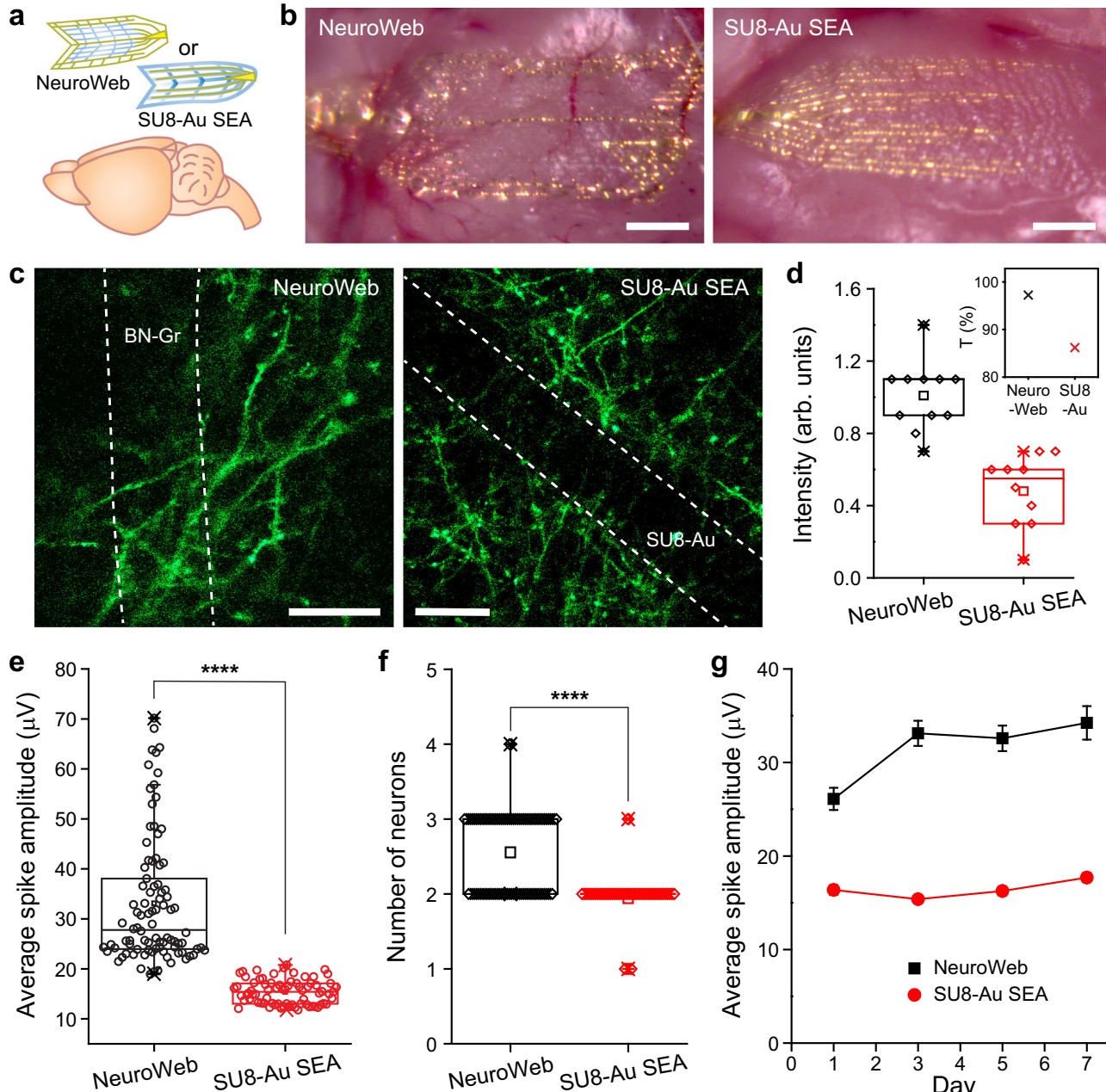

**Fig. 4 | Enhanced optical properties and electrophysiological capabilities of NeuroWeb. a** Schematic for transferring either NeuroWeb or SU8-Au SEA onto the mouse brain. **b** Optical images of NeuroWeb (left) and SU8-Au SEA (right) on the mouse brain. Scale bars, 1 mm. **c** Confocal in vivo imaging of myelinated axons in the transgenic mouse brain with NeuroWeb (left) and SU8-Au SEA (right) at day 1 after surgery. White dashed lines indicate the positions of BN-Gr (left) and SU8-Au ribbons (right). Scale bars, 30 μm. **d** Boxplots of the fluorescence intensities of myelinated axons underneath the BN-Gr (black; $N = 2$) and SU8-Au ribbons (red; $N = 2$), which were normalized by the intensities just outside the corresponding ribbons. The inset indicates the measured transmittance of NeuroWeb (black, 97.2%) and SU8-Au SEA (red, 86.2%), when the wavelength and spot size of the injected light were 488 nm and 100 μm, respectively. **e** Average spike amplitudes of NeuroWeb (black) and SU8-Au SEA (red) from 90 and 73 channels, respectively, at day 3 after surgery. ****: $P = 8.34 \times 10^{-5}$ (two-sided $t$-test). **f** Number of recorded single neurons per channel for NeuroWeb ($N = 90$) and SU8-Au SEA ($N = 73$). ****: $P = 2.26 \times 10^{-15}$ (two-sided $t$-test). Boxplots show mean (squares), median (horizontal lines), quartiles (boxes, 25–75%), and range (whiskers, 1–99%). **g** Average spike amplitudes of the detected action potentials across 1 week for NeuroWeb (black) and SU8-Au SEA (red). Eight mice ($N = 4$, each) used for statistical analysis. Error bars denote s.e.m. Source data are provided as a Source Data file.

To identify the neural signals recorded in the non-stimulated brain region, we analyzed signals in NeuroWebs A and B during separate optical stimulation of the neurons of Cb and S1 (Fig. 5f and g). First, the evoked spikes were observed in NeuroWeb B on the Cb (red trace, Fig. 5f) alongside the laser pulses incident on the Cb, while the responded spikes were detected a few milliseconds later in NeuroWeb A on the S1 (black trace, Fig. 5f). Next, when the S1 was optically stimulated, the evoked and responded signals were also observed in

NeuroWebs (Fig. 5g): NeuroWeb A on the S1 (black trace) and NeuroWeb B on the Cb (red trace) detected evoked spikes and few milliseconds delayed spikes, respectively, although the delayed times were slightly different from those observed with Cb stimulation. This finding illustrates that the S1 and Cb communicate with each other via neural signal transmission, depending on the pathway.

A quantitative analysis was performed using the time difference between the evoked and responded signals recorded in NeuroWebs A

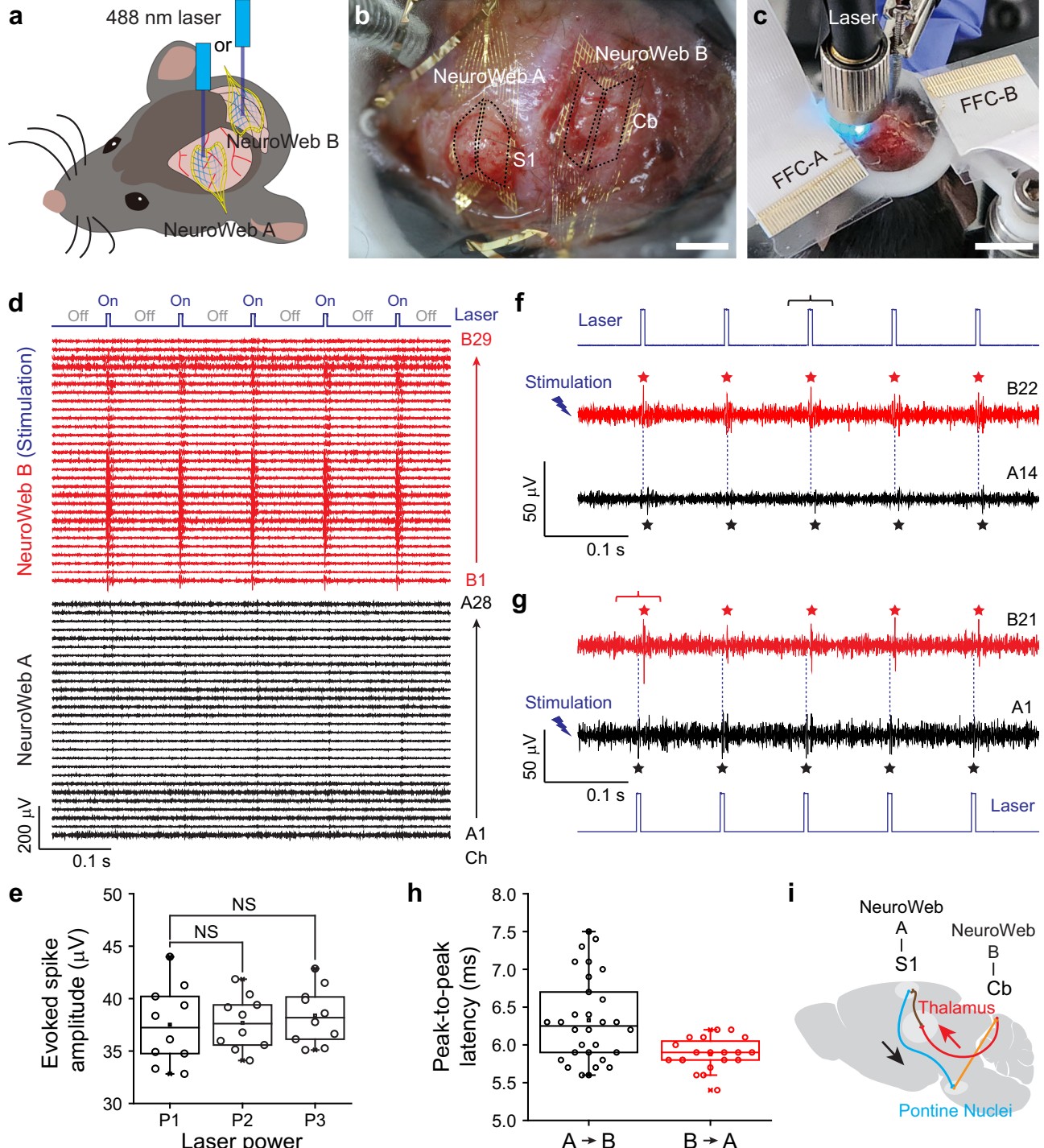

**Fig. 5 | Investigation of the neural connections between the somatosensory cortex (S1) and the cerebellum (Cb). a** Schematic of the optogenetics experiment. NeuroWebs A and B are placed on the surfaces of S1 and Cb, respectively, with one of them stimulated by a 488 nm laser. **b** Optical microscope image of the S1 and Cb of a live transgenic mouse with NeuroWebs A and B. Scale bar, 1 mm. **c** Photograph of optical stimulation to the brain of a live mouse using a fiber-coupled laser diode (pulse width: 5 ms, period: 10 Hz). Two NeuroWebs with 32 Pt electrodes are electrically connected to FFC-A and FFC-B via I/O pads. Scale bar, 1 cm. **d** Band-pass filtered (250–6000 Hz) extracellular spike recordings from the S1 (NeuroWeb A, black traces) and the Cb (NeuroWeb B, red traces), when the Cb is optically stimulated. The mouse is under anesthesia during the recording. The blue line indicates the voltage pulses applied to the laser diode (top). **e** Spike amplitudes recorded in NeuroWeb A with increasing laser power densities (P1: 1.30, P2: 1.43,

and P3: 1.59 mW/mm²). Boxplots show mean (squares), median (horizontal lines), quartiles (boxes, 25–75%), and range (whiskers, 1–99%). $P = 0.90$ and $P = 0.55$ (two-sided $t$-test). NS: not significant. Three mice were used for statistical analysis. **f** and **g** Representative band-pass filtered extracellular spike recordings from the S1 (NeuroWeb A, black traces) and the Cb (NeuroWeb B, red traces). A pulsed laser (blue line) optically stimulates the Cb (**f**) or S1 (**g**). The dotted lines are added to display the time difference between the evoked and responded spikes (★). The magnified views of the black and red brackets are shown in Supplementary Fig. 23. **h** Peak-to-peak latency between the evoked and responded spikes in the pathways from the S1 to Cb (A → B, black dots) and from the Cb to S1 (B → A, red dots). Boxplots show mean (squares), median (horizontal lines), quartiles (boxes, 25–75%), and range (whiskers, 1–99%). **i** Schematic of the neural circuits between the S1 and Cb[41]. Source data are provided as a Source Data file.

and B. The peak-to-peak latency of signals transmitting from the S1 to Cb ($N = 3$ mice) and from the Cb to S1 ($N = 2$ mice) was $6.33 \pm 0.50$ and $5.88 \pm 0.20$ ms, respectively (Fig. 5h). Therefore, the average time difference of ~0.45 ms indicates that neural signal transmission took slightly longer in the pathway from the S1 to Cb via the pontine nuclei than in the pathway from the Cb to S1 via the thalamus (Fig. 5i).

This finding was made possible by placing dual NeuroWebs in specific regions of a small live mouse brain and making tight contacts even on corrugated surfaces such as the Cb, allowing for excellent electrical detection of neural activities with a superior temporal resolution of 0.05 ms. Additionally, optical stimulation or imaging was successfully achieved using transparent NeuroWebs, without being obstructed by the body of the device. Furthermore, due to the light-weight and compact interface of the device, additional NeuroWebs were effectively added for brain mapping with no difficulties in synchronization of the signals detected by NeuroWebs (Supplementary Fig. 25).

## Discussion

In conclusion, our NeuroWeb is, to the best of our knowledge, the thinnest neural probe ever developed[15,17,27] and has demonstrated unique features in flexibility, adhesion and transparency. NeuroWeb exhibited remarkable electrophysiological performance comparable to implantable probes, revealing single-unit spikes with high amplitudes and SNRs without penetrating brain tissues, due to the ultra-thin thickness and rational design of the open lattice structure. We also proposed an efficient approach of vertical stacking to multiplexing to increase electrode density while maintaining device flexibility without relying on lithographic resolution. Furthermore, the exceptional transparency of NeuroWeb (nearly 100% transmittance) and strong adhesion were fully exploited in order to discover the neuronal correlation between the Cb and S1 via optogenetics experiment.

A single Neuroweb device can also be used to devote part of the electrodes to the recording and part to stimulate a specific population of cortical cells. We conducted an electrical stimulation experiment as a proof of concept (Supplementary Fig. 26). We fabricated 8 stimulation Pt electrodes with a radius of 50 µm and 24 recording Pt electrodes with a radius of 10 µm (Supplementary Figs. 24a and b). I/O pads were connected to FFC for internal electrical stimulation and recording, after surgery to place NeuroWeb in the S1 (Supplementary Fig. 26c). The evoked signals were then recorded using the Intan recording system with the amplifier. A representative trace shows the response to electrical stimulation (Supplementary Fig. 26d). In the firing rate of this channel, neural activity more than doubles after the stimulation (Supplementary Fig. 26e).

Therefore, NeuroWeb can open a next-generation paradigm in minimally invasive brain mapping probes by (1) enabling chronic manipulation and monitoring of single-unit neural activities without implantation of the probe into the deep brain tissue, (2) creating multiplexed recording sites without significantly increasing stiffness, and (3) allowing optical and electrophysiological mapping with high spatial and temporal precision, using its transparency, to obtain information of complex neural activities from multiple regions of the brain. In addition, NeuroWeb can provide superior long-term performance without the phototoxic side effects commonly associated with full-optical approaches, such as those utilizing calcium- or voltage-sensitive dyes[46,47]. Furthermore, NeuroWeb allows for electrophysiological mapping via internal electrical stimulation as well as external optical stimulation, further expanding its functionality and applications in neuroscience research.

## Methods

### Vertebrate animal subjects

Adult (8-week) female CD-1 mice (Charles River Laboratories, Wilmington, MA) were used as vertebrate animal subjects in this study for brain surface electrophysiology and whiskers stimulation. The Thy1-EGFP line M (Jackson Labs #007788) mice (3-month) were used as vertebrate animal subjects for in vivo confocal fluorescence imaging. Adult (8-week) male Thy1-ChR2-YFP mice (Jackson Laboratory) were used as vertebrate animal subjects for the optical stimulation of genetically modified neurons and electrophysiological recording. All animal experiments were approved by the Korea University Institutional Animal Care & Use Committee (KUIACUC2019-0024).

### Fabrication of NeuroWeb

The fabrication steps for NeuroWeb are as follows: (1) a 3″-diameter $SiO_2$/Si wafer (Boron-doped p-type Si of 1–5 Ω·cm resistance, 280 nm-thick $SiO_2$; Silicon Materials Inc.) was pre-cleaned with acetone and isopropanol and dried in an $N_2$ flow. A 100 nm-thick Ni layer was deposited as a sacrificial layer on the $SiO_2$/Si substrate using a thermal evaporator (Thermal evaporator; NNS Vacuum) at a high vacuum of $<1 \times 10^{-6}$ Torr. (2) Lift-off resist (LOR 3 A; Kayaku Advanced Materials) was spin-coated on the Ni sacrificial layer (4000 rpm for 30 s) and baked at 180 °C for 3 min. Positive photoresist (PR, Microposit S1805; Shipley) was spin-coated on LOR 3 A (4000 rpm for 30 s) and baked at 110 °C for 80 s. The PR/LOR 3 A on the Ni sacrificial layer was selectively exposed to UV with an energy density of 150 mJ/cm² using a mask aligner (MDA-400N; Midas System). The wafer was immersed into the developer (AZ 300 MIF developer; Merck) for 50 s, rinsed with deionized (DI) water, and dried with an $N_2$ flow. The wafer was subsequently deposited with a 5 nm-thick Cr adhesion layer and a 50 nm-thick Pt layer using electron-beam evaporation (E-beam evaporator; ULVAC) at a high vacuum of $<1 \times 10^{-6}$ Torr, followed by removal of the Pt-coated resist via lift-off (mr-Rem 700; Microresist) at 60 °C for 1 h. These processes resulted in the formation of 32-channel bottom Pt electrodes with a diameter of 20 µm on the Ni sacrificial layer. (3) Step (2) was repeated, but in this step, a 100 nm thick Au layer was deposited using a thermal evaporator on the Ni sacrificial layer to form I/O pads. (4) The SU-8 layer was patterned to identify the position of the active region of NeuroWeb. The SU-8 (SU-8 2000.5; MicroChem Corp.) was spin-coated at 2000 rpm for 30 s, and baked at 65 °C and 95 °C for 1 min. After exposing SU-8 to UV with an energy density of 600 mJ/cm², the wafer was post-baked at 65 °C for 1 min and 95 °C for 3 min. The SU-8 was developed (SU-8 developer; MicroChem Corp.) for 1 min, rinsed with isopropanol, dried with an $N_2$ flow, and hard-baked at 190 °C for 2 h. (5) CVD-grown multilayer h-BN (CVD multilayer h-BN on a copper foil; 2D Semiconductors), acting as a bottom passivation layer, was transferred to the active region of NeuroWeb using the PMMA wet transfer method (PMMA C4; MicroChem Corp.). (6) Step (2) was repeated to pattern PR on the bottom of the multilayer h-BN to allow the usage of an etching mask. (7) $O_2$ plasma etching ($O_2$ plasma etcher; JP Technologies Co., Ltd.) with a power of 50 W and 20-sccm $O_2$ was conducted for 55 s, to define the pattern of the transferred h-BN layer using S1805 PR mask. The h-BN was partially etched in this step and fully disappeared when the freestanding structure of NeuroWeb was formed. The remaining PR was removed by immersing it in a developer (AZ 300 MIF developer; Merck) for 1 min, after exposing it to UV with an energy density of 600 mJ/cm². The wafer was then rinsed with DI water and dried in an $N_2$ flow. (8) Step (4) was repeated to fabricate the bottom SU-8 to be used as a bottom passivation layer for metal interconnects and as a supporting backbone for the supporting region. (9) Step (3) was repeated to pattern Au interconnects on the bottom SU-8 layer. Au interconnects electrically connect Gr and I/O pads and are also used for the supporting region. (10) Step (5) was repeated to transfer three layers of Gr (CVD monolayer Gr on a copper foil; Graphene Supermarket) to the active region using the PMMA wet transfer method. (11) Step (2) was repeated to pattern PR on three layers of Gr for the usage of an etching mask. (12) Step (7) was repeated to pattern Gr interconnectors. (13) Step (5) was repeated to transfer multilayer h-BN to the active region of NeuroWeb, acting as a top passivation

layer. (14) Step (4) was repeated to pattern SU-8 for the usage of an etching mask. (15) Step (7) was repeated to define the pattern of the top h-BN layer. (16) Step (4) was repeated to pattern the SU-8 layer acting as the top passivation layer for Au interconnects. (17) Step (2) was repeated to fabricate the top Pt electrodes, resulting in double-sided recording electrodes. (18) The Ni sacrificial layer was wet-etched in a solution of 40% FeCl₃: 39% HCl: H₂O (1:1:20) to release NeuroWeb from the SiO₂/Si wafer.

## Evaluating the characteristics of NeuroWeb

First, after fabricating the NeuroWeb on the wafer, we use optical microscopy to verify that the Gr and Au interconnectors are properly connected. Then, the Ni sacrificial layer is etched to create a free-standing structure of NeuroWeb, and the impedance of a randomly selected NeuroWeb is measured to validate the device characteristics in the same batch. Finally, we measure the impedance after surgery to ensure the structural integrity of NeuroWeb.

Normalizing impedance values per area, double-sided Pt electrodes with a 20 μm diameter have a value of ~$339 \times 10^3$ kΩ μm², which is comparable to the value in previous work[48]. In addition, in the measured EIS, the double-sided Pt electrode has a lower impedance at all frequencies than the single-sided Pt electrode[48].

## Fabrication of SU8-Au SEA

Key steps involved in the fabrication of SU8-Au SEA are described as follows: (1) a 3″-diameter SiO₂/Si wafer was pre-cleaned and coated with a sacrificial Ni layer. (2) Bottom Pt electrodes were fabricated by a series of pattering, deposition, and lift-off processes. (3) Negative photoresist SU-8 was applied and patterned on the SiO₂/Si wafer using photolithography for bottom SU-8 passivation. (4) Another series of pattering, deposition, and lift-off processes were performed to fabricate the Au interconnect lines on the bottom SU-8. (5) Step (2) was repeated to fabricate top Pt recording electrodes. (6) Step (3) was repeated to fabricate the top SU-8 layer, which insulates the metal interconnect lines, while leaving the Pt recording electrodes exposed[30].

## Analysis of NeuroWeb's dimensions

The use of 2D materials such as Gr and h-BN does not restrict device dimensions. In terms of material preparation, the CVD growths of Gr and h-BN allow for the fabrication of materials with dimensions in the range of tens of centimeters. These materials have well-developed transfer methods that allow for the creation of large electrode arrays[49,50]. In addition, NeuroWeb employs conventional photolithography techniques, enabling the fabrication of devices without the size limitations of electrode arrays. Moreover, in terms of handling, NeuroWeb is primarily managed in a PBS solution with a pipette. Because NeuroWeb is placed on the surface of the brain as opposed to being implanted in the deep brain, the pipette dimensions do not restrict NeuroWeb's size.

The top SU-8 layer in the active region increases the mechanical strength of the active region. Using only h-BN lacks sufficient mechanical strength to maintain the shape of the active region in a freestanding structure. On the other hand, when using SU-8 alone, a difference in film thickness of about 5–15% is unavoidable due to film edge buildup, surface tension, and the shape and size of the substrate. When considering the thickness of the SU-8, this 5–15% difference corresponds to a change in thickness ranging from tens to hundreds of nanometers.

## Flexibility and adhesion strength experiments with NeuroWeb

Flexibility experiment: Periodic SU-8 line patterns with a width of 10 μm and height of 5 μm were fabricated on the Au-coated substrate at various spacing distances (d) from 50 to 80 μm. After transferring NeuroWeb and SU8-Au SEA to the substrate, the resistance between the electrode and the Au-coated substrate was measured (2450 SourceMeter; Keithley).

Adhesion strength experiment: NeuroWeb and SU8-Au SEA in DI water were transferred to the slide glass (Microscope slides; Marienfeld) using a glass pipette, and left to dry naturally. The slide glasses were fixed to plastic petri dishes using Scotch tape. DI water was poured over the probes on the slide glass until they were completely immersed. Then, we observed the probes on the slide glass using zoom lens (Zoom 7000 Macro Lens; Navitar) and CCD camera (DCU224C CCD Camera; Thorlabs). The results show that, once attached to a surface, NeuroWeb cannot be detached without suffering damage.

## Electrical characterization

In vitro impedance measurements of NeuroWeb and stacked NeuroWeb were performed using the Intan system at 1 kHz (Supplementary Fig. 4a) and 20 Hz to 4 kHz (Supplementary Fig. 4b), after immersing the active region of each probe in 1× PBS. Resistance in Fig. 2e was measured using a source measure unit (2450 SourceMeter; Keithley) and a customized probe station, by probing the I/O pad of NeuroWeb and Au-coated substrate with a voltage range from −1 V to 1 V. A small Gr resistance of ~47 kΩ, which is the average resistance of NeuroWeb's Gr line, is not having a detrimental effect on the electrode impedance.

## AFM imaging

AFM imaging (XE-150; Park Systems Inc.) was utilized to measure the thickness of a single SU-8/h-BN/Gr/h-BN ribbon in NeuroWeb prior to etching the Ni layer. A non-contact cantilever (PPP-NCHR; Park Systems Inc.) was employed as a probe in the non-contact mode. The measured area was 50 μm × 50 μm, while the scan rate was 0.5 Hz at a resolution of 256 × 256 pixels.

## Numerical simulations

Using the finite-element method (FEM; COMSOL Multiphysics), we performed numerical simulations to calculate the total adhesive elastic stress on an h-BN thin film adhered on a biomaterial substrate. h-BN films of various thicknesses ranging from 60 to 1000 nm were examined. The top SU-8 layer was not considered in the simulation for simplicity. Periodic patterns with 5 μm height, 10 μm width, and 60 μm periodicity were introduced on top of the substrate with a total length of 400 μm. The shape of the h-BN film was determined using the SEM images, as shown in Supplementary Fig. 6b. The angle θ between the h-BN film and the substrate was assumed to decrease linearly with increasing h-BN thickness; θ was set to 30° for the 60 nm-thick h-BN and 0° for the 1000 nm-thick h-BN. Young's modulus and Poisson's ratio of h-BN were 865 GPa and 0.22, respectively[51]. Young's modulus and Poisson's ratio of the substrate were 620 Pa and 0.45, respectively[52].

At the beginning of the simulation, the left 120 μm of the h-BN film was separated from the substrate. The total adhesive elastic energy for the entire structure was calculated using the Cohesive Zone Model (CZM)[53] in the FEM simulation, when the remaining h-BN was completely separated from the substrate. The total adhesive elastic stress on unit volume was obtained by dividing the total adhesive elastic energy by the total volume of the h-BN thin film. The von Mises stress was used to display the calculated images.

## Surgical steps using NeuroWeb

Mice were anesthetized using an inhalational anesthesia system (Single Animal Isoflurane Anesthesia System; Harvard apparatus) based on vaporized isoflurane (Isotroy isoflurane USP; Troikaa Pharmaceuticals Ltd.) with oxygen gas (3% flow rate of isoflurane with 1.0 L/min O₂). Surgery was started after anesthesia was confirmed by no response to a front toe pinch. After confirmation, the flow rate of isoflurane was reduced to 2% and maintained until the surgery was finished. During the surgery, a thermal control pad (Far Infrared Warming Pad; Kent

Scientific Corp.) was used to keep the mouse body temperature at 37 °C. A sterile eye lubricant (Puralube Veterinary Pomade Eye Ointment; Dechra) was applied to the mice eyes to prevent corneal damage and dryness. A 1 cm-long incision was made along the sagittal sinus, and skin was resected to expose an area of approximately 1 cm × 1 cm using surgical scissors and a scalpel that had been sterilized by an autoclave (Steam Sterilizer; Sejong Science) for 15 min at 121 °C, 15 psi, and 70% ethanol, followed by rinsing with sterile 1× PBS solution. A dental drill (EXL-M40 dental drill; Osada Inc.) was used to drill two 1 mm burr holes in front of the bregma, and sterilized set screws (stainless steel set screws with a groove diameter of 1 mm) were inserted into each of these holes to serve as a point of fixation for the head stage as well as a grounding and reference electrode. The customized poly-lactic acid (PLA) head stages were fabricated using a 3D printer (Ender 6; Creality) and sterilized with 70% ethanol. The stage was prepared by mounting a sterile FFC folded in an L-shape on a piece of sterile polyvinyl chloride (PVC) film fixed to the head stage using dental cement (Superbond C&B dental cement; Sun medical). The head stages were also fixed to the skull and mounting screws using dental cement. The hole with an area of ~3 mm × 3 mm was created in the skull using dental drill. The dura was removed carefully, and a PBS solution was applied to wet the mouse skull prior to insertion of NeuroWeb. NeuroWeb stored in PBS solution was drawn into a glass pipette, and the active region of NeuroWeb was then transferred to the mice brain surface. The I/O pads made electrical connection with FFC using a direct-contact method[33]. The active region of NeuroWeb was covered with a 3 mm-diameter round glass and secured with dental cement. The I/O pad interface was covered with a piece of PVC film and sealed with dental cement and quick drying adhesive (Loctite Epoxy Adhesive; Henkel). The mice were monitored during recovery.

### In vivo confocal imaging of mice brain surface
We used 3-month-old Thy1-EGFP line M (Jackson Labs #007788) mice to observe in vivo fluorescence images. After NeuroWeb was attached to the brain surface, a sterile round coverslip with a diameter of 5 mm (#1 thickness; Warner Instruments) covered the exposed area of the skull with dental cement. A custom-made metal plate was mounted to the mice head with dental cement to prevent vibration during imaging. The mice were then anesthetized with isoflurane (1.5% in oxygen to maintain a breathing frequency of around 1–2 Hz) and placed on a 3D motorized stage (MPC-385; Sutter Instrument) heated by a heat blanket at 37–38 °C. We used a customized confocal microscope with a water immersion objective lens (60× Nikon CFI APO NIR Objective, 1.0 NA, 2.8 mm WD)[40]. As shown in Supplementary Fig. 15a, a 488 nm laser (central wavelength, 488 nm; bandwidth, 1.5 nm; 06-MLD; Cobolt) was used for excitation, and coupled with a scanning system based on galvanometer scanning mirrors (6220H; Cambridge Technology). The fluorescence signal was collected by PMT (PMTSS2; Thorlabs) after passing through a dichroic filter (505–800 nm transmission, MD498; Thorlabs), whereas the reflected signal was collected by PMT (PMTSS2) after being reflected by a dichroic filter (452–490 nm reflection, MD498; Thorlabs). The fluorescence emission images and reflection images were acquired at a depth interval of 1 µm in the cortical layer 1 of all the mice. The field of view of each image was 100 µm. The images were analyzed using MATLAB (MathWorks).

### In vivo brain surface recordings in mice
The average noise level of the recorded signals in PBS was $2.20 \pm 0.45$ µV, which is within a reasonable range for neural recording applications[54]. Then, mice brain surface recordings with NeuroWeb were obtained at 1, 3, 5, and 7 days after surgery. Mice were restrained in a Tailveiner restrainer (Braintree Scientific LLC) to obtain restrained recordings. An Intan evaluation system (Intan Technologies LLC) with a 20 kHz sampling rate and a 60 Hz notch filter was used for electrophysiological recording.

### Analysis of electrophysiological recording data
The electrophysiological recording data were analyzed offline. Original recording data with NeuroWeb were filtered using noncausal Butterworth band-pass filters ("filtfit" function in Matlab) in the 250–6000 Hz frequency range to extract single-unit spikes[8]. Single-unit spike sorting was performed by amplitude thresholding, and the WaveClus software was used to identify the number of recorded single neurons. Spikes assigned to the same cluster were coded with the same color.

### Optical stimulation and electrophysiological recording
A 488 nm laser diode (LP488-SF20G 488 nm/20 mW; ThorLabs, Inc.) was used for optical stimulation. The laser diode was connected to a function generator (33521 A; Agilent) to apply voltage pulses with a frequency of 10 Hz and a pulse width of 5 ms. To electrically identify optical stimulation, the function generator was synchronized with an Intan recording system. The output of the laser diode was coupled to an optical fiber to make a uniform beam shape.

### Estimation of the number of stimulated neurons in optogenetics experiments
To estimate how many cortical layers of cells are stimulated during a pulse, we assume that only cells receiving blue light with a power density greater than the activation threshold of channelrhodopsin-2 (ChR2) (1 mW/mm$^2$) are stimulated[55]. Because the output power densities of the pump laser were 1.30, 1.43, and 1.59 mW/mm$^2$, and the light intensity ratio exponentially decreases with the depth of brain tissue[56,57], the stimulated depths are estimated to be ~0.02, ~0.03, and ~0.04 mm, respectively. This implies that only the first cortical layer in the mouse brain was optically stimulated[58].

Next, to estimate the number of neurons stimulated during the process, we calculated the volumes of the stimulated regions by multiplying the pumping area (~0.5 mm$^2$) by the stimulation depths indicated above: 0.01, 0.015, and 0.02 mm$^3$. Since the neuronal cell densities in the first cortical layer of S1 and the molecular layer of Cb are 29,213 and 14,000 cells per mm$^3$, respectively[59,60], we estimate the number of stimulated neurons to be 292, 438, and 584 cells in the S1 and 140, 210, and 280 cells in the Cb, for laser power densities of 1.30, 1.43, and 1.59 mW/mm$^2$, respectively.

### Statistical analysis
Matlab, Origin, Python, and Excel were applied to all statistical analyses. In the figure captions, all replicate numbers, error bars, P values, boxplots, and statistical tests were specified. For experiments demonstrating statistical significance, sample sizes were selected to achieve at least 80% power at an alpha level of 0.05.

### Reporting summary
Further information on research design is available in the Nature Portfolio Reporting Summary linked to this article.

## Data availability
All the data supporting the findings of this study are available within this paper and its Supplementary Information. Any additional information can be obtained from the corresponding author on request. Source data are provided with this paper.

## Code availability
The MATLAB codes used in this work are available from the corresponding author upon request.

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

## Acknowledgements

This work was supported by the National Research Foundation of Korea (NRF) grant funded by the Korean government (MSIT) (2021R1A2C3006781), the Samsung Research Funding and Incubation Center of Samsung Electronics (SRFC-MA2001-01), and the New Faculty Startup Fund from Seoul National University. J.H.H., Y.J. and W.C. acknowledge support from the Institute for Basic Science (IBS-R023-D1).

## Author contributions

J.M.L. and H.G.P. designed the experiments. J.M.L., Y.W.P., Y.J. and J.H.H. performed the experiments. J.M.L. and Y.J.K. performed the simulations. J.M.L., Y.W.P., W.C., D.L. and H.G.P. analyzed the data. J.M.L., Y.W.P. and H.G.P. wrote the paper. All authors discussed the results, revised or commented on the paper. H.G.P. supervised the project.

## Competing interests

The authors declare no competing interests.
