## [Peer Review File · Nature Communications]

REVIEWER COMMENTS

Reviewer #1 (Remarks to the Author):

The manuscript of J.M. Lee et al., reports on a very unique and very interesting design and utilization of a new neural probe – NeuroWeb. The authors describe their probe as atomically thin, which is quite a stretch, considering the smallest thickness of the probes is $\sim 100\text{nm}$ at the lowest end. Perhaps nanoscale would be most appropriate. Nonetheless, getting to the 100nm thickness is a fabulous achievement! Besides, authors did use atomically thin materials in the work such as graphene and hBN, yet in a rather unorthodox manner, with using graphene as feedlines instead of the interface material. This is typically unique, and authors do report on the novel technology that is noteworthy and I would strongly suggest publishing the work in Nature Communications. The work is original, well supported by the data and experiments, and methodology is sound. I do have a few queries/comments as described below to ensure better comprehension of the manuscript, mainly focusing on a few technical points:

- From the very beginning, the text and Figure 1, it is not entirely clear which regions of the probes are $\sim 100\text{nm}$ thick, and which are actually slightly more classic, $5\text{-}10\mu\text{m}$. This issue has been bothering me until a few rounds of back-and-forth in the manuscript itself to understand that the probe consists of two kind of separated area – the large “classic” probe with the inner window consisting of much thinned hBN-graphene-hBN probe. I suggest authors to clarify this as early as possible to avoid confusion. This could be addressed with some on-figure drawings (not the caption) over Fig. 1e-f.
- The paper shows a rather very unusual use of graphene and gold. Why not otherwise? Gold is more conductive, graphene is transparent and makes better interface. Why not fully graphene or hBN/graphene/hBN?
- I would like to further follow-up on the claim of hBN-graphene-hBN use in the “active window”, which has been claimed through the manuscript. In the Figure 1b and later in the AFM scans, however, it is shown that there is an additional SU-8 on top of the hBN. Kindly elaborate and clarify on the matter.
- Now, if there is SU-8, what is the point of SU-8 on top of hBN, or vice versa? Can the authors elaborate on the use of just either one of them, and compare directly? One may simply use SU-8 or only hBN. Why both? Is there an advantage that is supported by the experimental data?
- Further on the topic, when the authors describe the COMSOL-based calculations, they only consider hBN: “The simulation showed that adhesive stress decreased rapidly with increasing h-BN thickness (Fig. 2g), consistent”. Why only hBN, if there is SU-8, which is actually the main interface?
- Regarding the cross-channel conductivity data. The Data shown in Fig. 1c is nice, but seems rather premature and such study needs more than just current, the authors rather need to study EIS cross-sensitivity. If a channel is not open, but a pulse is delivered through, will it be picked up?... It might be considering such a tiny passivation layer. Comprehensive EIS study of the cross-reactivity is required here...

- Besides, seems like authors register $\sim 15\text{E-}6\text{A}$ current when applying 1V potential; this means the channel/feedline resistance is $R=V/I=1\text{V}/15\text{E-}6=\text{approx } 66.6\text{ kOhm}$, which is rather significant.
- On line 134, the authors claim that “There was no current leakage even when a PBS solution was put on top of the h-BN layer” – this is not seem in the linear plot. Please provide the number? nA? pA? what was the voltage applied? Kindly show the plot in log scale, and/or provide the actual values of the current/resistance measured.
- On line 158, the authors mention “The average impedance of $\sim 540\text{ k}\Omega$ is...”. Can you please normalize this per area and compare to the state-of-the-art? Now this must be interesting; would you normalize per $(\pi*r^2)$ with r being the diameter, or $(2*\pi*r^2)$, considering your electrodes are opened from both sides? Which configuration was the EIS performed? Have you performed EIS before/after resuspension perhaps to see that your impedance decreases when you open up the second part of the electrode?
- On line 305, the authors claim “NeuroWeb shows superior performance comparable to implantable probes, while causing no brain tissue damage.”. Now this might be an overstatement. How would you safely remove those? Do you leave them inside? Then you must study long-term effect before claiming "no tissue damage".
- In Methods (line 671), authors claim that $\sim 30\text{nm}$ thick hBN is etched away in 50W oxygen plasma in 55 seconds, which seems rather unlikely hBN should (a) not be etched away by simple O2 plasma; definitely not 30nm of it, in MILD plasma for only 55 seconds... At very harsh plasma conditions, the max rate found is $\sim 10\text{-}20\text{nm}/\text{min}$ [10.1021/acsphotonics.8b00127]. Kindly confirm details of this process step.
- Finally, I wonder on how do the authors ensure structural integrity of the NeuroWeb while handling the probes? After fabrication and releasing, during EIS/characterization, and during/after implantation? It seems a very technical details but potential paramount to understanding of the probe’s potential.

Reviewer #2 (Remarks to the Author):

In this paper, the authors report an atomically thin, minimally invasive neural probe ("NeuroWeb") composed of hexagonal boron nitride and graphene, which combines the advantages of SEA and iMEA. The authors have realized the open-lattice structure of the NeuroWeb with very thin electrodes only 100 nm thick. The NeuroWeb exhibits high flexibility and strong adhesion, and various experiments have demonstrated that it can be applied to uneven mouse brain surfaces and conformal and tight interfaces. In particular, in in vivo electrophysiological recording experiments, the authors' NeuroWeb detects stable single-unit activity of neurons with a high signal-to-noise ratio. Furthermore, taking advantage of its thinness and high transparency, the authors have successfully combined light stimulation and measurement techniques with electrical measurement techniques. They actually investigated the neural interaction between the somatosensory cortex and the cerebellum using light stimulation and measured the neural signal transmission time between brain regions according to pathways. I acknowledge that

the novel thin-film brain electrode NeuroWeb proposed by the authors is an important method to better understand complex brain networks through optical and electrophysiological mapping of the brain. I believe that the novelty of this paper as a brain electrode device and the accuracy of the brain measurements are worthy of publication in Nature Communications, but I would like the authors to answer a few questions before publication to make the paper better. I describe them as follows.

1. A major issue is the susceptibility to noise when the impedance of the measurement probe and its contact impedance to the brain is high. The authors' brain electrodes have great features because they are very thin, but on the other hand, the overall impedance appears to be high. For example, the impedance shown in Extended Data Fig. 4. is several hundred kilo-ohms or more. I believe that the normal impedance for the brain is around a few kilo-ohms. If the impedance is two orders of magnitude higher, the signal-to-noise ratio will be much worse.

Please create a quantitative comparison table with the normal brain probes already reported by other groups to quantitatively demonstrate the usefulness of the NeuroWeb. If NeuroWeb is able to achieve a higher signal-to-noise ratio than other existing probes despite its high impedance, and if NeuroWeb is able to measure cleaner brain signals than conventional probes, please explain the reason in more detail. It is generally known that probes with high impedance are vulnerable to disturbance noise and crosstalk between wires.

2. Measuring electrophysiological responses when the brain is stimulated is a very important experiment in the development of brain electrodes. I think that the experimental results shown in Fig. 5 by the authors are excellent. On the other hand, I would like to see the results of Figures 5f and g in more detail, so please add a magnified view. It is important to show directly in the figure what the waveform looks like and after how many seconds it reacts.

3. For high impedance or thin film electrodes, it is expected that their characteristics will vary greatly with each frequency. Please show the figure with frequency on the horizontal axis and impedance on the vertical axis. If the characteristics differ significantly from frequency to frequency, it will be difficult to measure over a wide frequency range.

4. It is very important for brain research that the electrodes and wiring are transparent and can be linked to the optical system. On the other hand, the figure does not show experimental results that demonstrate quantitative performance regarding transparency. I would like to request the addition of a figure that quantitatively shows transparency as frequency on the horizontal axis and transparency on the vertical axis, as well as a discussion of this issue.

Response to Reviewer #1.

Comment. The manuscript of J.M. Lee et al., reports on a very unique and very interesting design and utilization of a new neural probe – NeuroWeb. The authors describe their probe as atomically thin, which is quite a stretch, considering the smallest thickness of the probes is ~100nm at the lowest end. Perhaps nanoscale would be most appropriate. Nonetheless, getting to the 100nm thickness is a fabulous achievement! Besides, authors did use atomically thin materials in the work such as graphene and hBN, yet in a rather unorthodox manner, with using graphene as feedlines instead of the interface material. This is typically unique, and authors do report on the novel technology that is noteworthy and I would strongly suggest publishing the work in Nature Communications. The work is original, well supported by the data and experiments, and methodology is sound. I do have a few queries/comments as described below to ensure better comprehension of the manuscript, mainly focusing on a few technical points:

Our response. We thank Reviewer #1 for his/her positive evaluation of the importance of our work. We are happy to have the opportunity to address the reviewer’s critical remarks, important questions, and specific suggestions.

Comment 1. From the very beginning, the text and Figure 1, it is not entirely clear which regions of the probes are ~100nm thick, and which are actually slightly more classic, 5-10 μ m. This issue has been bothering me until a few rounds of back-and-forth in the manuscript itself to understand that the probe consists of two kind of separated area – the large “classic” probe with the inner window consisting of much thinned hBN-graphene-hBN probe. I suggest authors to clarify this as early as possible to avoid confusion. This could be addressed with some on-figure drawings (not the caption) over Fig. 1e-f.

Our response. We apologize for confusing the reviewer. As the reviewer pointed out, the probe consists of two separated areas – a 1 μ m-thick outer region consisting of SU-8/Au/SU-8 (supporting region and metal interconnects) and a 100 nm-thick inner region consisting of SU-8/h-BN/Gr/h-BN (active region).

To clarify this point and avoid confusion, we revised Fig. 1e-f as the reviewer suggested. In addition, we added the following sentences to the revised manuscript (page 5): “**This active region consisting of SU-8/h-BN/Gr/h-BN is designed to have a thickness of 100 nm.**” and “**The supporting region and metal interconnects have a thickness of ~1 μ m.**”

[Revised Fig. 1e-f]

Comment 2. The paper shows a rather very unusual use of graphene and gold. Why not otherwise? Gold is more conductive, graphene is transparent and makes better interface. Why not fully graphene or hBN/graphene/hBN?

Our response. We thank the reviewer for raising this point. As stated in the original manuscript, the supporting region consisting of the SU-8/Au/SU-8 open lattice structure allows the active region (h-BN/graphene/h-BN) to spread out without tangling (page 5). In addition, Au joints were fabricated on Pt electrodes for a solid mechanical connection between the electrodes and h-BN ribbons (page 9).

To clarify this point, we added a sentence to the revised manuscript (page 5): “The active region is attached to the brain surface without wrinkles by leveraging the strength of the supporting region.”

Comment 3. I would like to further follow-up on the claim of hBN-graphene-hBN use in the “active window”, which has been claimed through the manuscript. In the Figure 1b and later in the AFM scans, however, it is shown that there is an additional SU-8 on top of the hBN. Kindly elaborate and clarify on the matter.

Our response. We thank the reviewer for raising this point. As stated in the original manuscript, the top SU-8 layer increases the mechanical strength of the active region (page 5). This SU-8 layer contributes to the overall thickness of 100 nm of the active region.

To clarify this point, we added a sentence to the revised manuscript (page 5): “**This active region consisting of SU-8/h-BN/Gr/h-BN is designed to have a thickness of 100 nm.**” In addition, we added a paragraph to the Methods section (‘Analysis of NeuroWeb’s dimensions’) in the revised manuscript (please see our response to Comment 4).

Comment 4. Now, if there is SU-8, what is the point of SU-8 on top of hBN, or vice versa? Can the authors elaborate on the use of just either one of them, and compare directly? One may simply use SU-8 or only hBN. Why both? Is there an advantage that is supported by the experimental data?

Our response. We thank the reviewer for raising this point. As stated in Comment 3, using only h-BN lacks sufficient mechanical strength to maintain the shape of the active region in a freestanding structure. On the other hand, when using SU-8 alone, a difference in film thickness of about 5% to 15% is unavoidable due to film edge buildup, surface tension, and the shape and size of the substrate [Coatings 11, 1322 (2021)] [J. Am. Chem. Soc. 126, 13778 (2004)]. When considering the thickness of the SU-8, this 5–15% difference corresponds to a change in thickness ranging from tens to hundreds of nanometers. Thus, we designed the device using thin SU-8 and h-BN.

To respond to the reviewer’s comment, we added a paragraph to the Methods section (‘Analysis of NeuroWeb’s dimensions’) in the revised manuscript: “**The top SU-8 layer in the active region increases the mechanical strength of the active region. Using only h-BN lacks sufficient mechanical strength to maintain the shape of the active region in a freestanding structure. On the other hand, when using SU-8 alone, a difference in film thickness of about 5% to 15% is unavoidable due to film edge buildup, surface tension, and the shape and size of the substrate. When considering the thickness of the SU-8, this 5–15% difference corresponds to a change in thickness ranging from tens to hundreds of nanometers.**”

Comment 5. Further on the topic, when the authors describe the COMSOL-based calculations, they only consider hBN: “The simulation showed that adhesive stress decreased rapidly with increasing h-BN thickness (Fig. 2g), consistent”. Why only hBN, if there is SU-8, which is actually the main interface?

Our response. We thank the reviewer for raising this point. SU-8 is a secondary structure in NeuroWeb that only exists on one side of the h-BN. Thus, only h-BN was considered in the simulation to calculate the total adhesive elastic stress of the device adhered on a substrate.

To clarify this point, we added a sentence to the Methods sections (‘Numerical simulations’) in the revised manuscript: “The top SU-8 layer was not considered in the simulation for simplicity.”

Comment 6. Regarding the cross-channel conductivity data. The Data shown in Fig. 1c is nice, but seems rather premature and such study needs more than just current, the authors rather need to study EIS cross-sensitivity. If a channel is not open, but a pulse is delivered through, will it be picked up?... It might be considering such a tiny passivation layer. Comprehensive EIS study of the cross-reactivity is required here...

Our response. We thank the reviewer for raising this point. As the reviewer suggested, we systematically measured cross-impedance values at 100 and 1000 Hz with and without PBS. The impedance measurement at 100 Hz is 2.4 M Ω without PBS and 2.2 M Ω with PBS, and at 1000 Hz, it is 2.3 M Ω without PBS and 2.2 M Ω with PBS. The measurement results show a slightly lower impedance with PBS in both cases, but the difference is not significant enough to affect the measurement; no frequency dependence is observed, as shown in the graph below.

[Fig. R1] Measured cross-impedance.

To respond to the reviewer’s comment, we added a sentence to the revised manuscript (page 8): “No frequency dependence was also measured.”

Comment 7. Besides, seems like authors register $\sim 15\text{E-}6\text{A}$ current when applying 1V potential; this means the channel/feedline resistance is $R=V/I=1\text{V}/15\text{E-}6=\text{approx } 66.6\text{ k}\Omega$, which is rather significant.

Our response. We thank the reviewer for raising this important point. The length of the graphene line in Fig. 1c ($5\ \mu\text{m} \times 3.1\ \text{mm}$) is larger than that of NeuroWeb, resulting in

higher resistance. Since three-layer graphene has a sheet resistance of $\sim 60 \text{ } \Omega/\text{square}$ [Nat. Nanotechnol. 5, 574 (2010)] [Nat. Photon. 4, 611 (2010)] [Nature 457, 706 (2009)], the resistance of NeuroWeb's longest graphene line ($5 \text{ } \mu\text{m} \times 2.8 \text{ mm}$) is estimated to be $\sim 33.6 \text{ k}\Omega$, and this value is similar to the measured average resistance in Fig. 2e ($\sim 47 \text{ k}\Omega$).

To clarify this point, we provide a more precise dimension of the graphene line in the caption of Fig. 1c: “ ~ 3 ” was changed to “**3.1**” in the revised manuscript. In addition, we added a sentence to the Methods section (‘Electrical characterization’) in the revised manuscript: “A small graphene resistance of **$\sim 47 \text{ k}\Omega$, which is the average resistance of NeuroWeb's graphene line**, is ...”

Comment 8. On line 134, the authors claim that “There was no current leakage even when a PBS solution was put on top of the h-BN layer” – this is not seen in the linear plot. Please provide the number? nA? pA? what was the voltage applied? Kindly show the plot in log scale, and/or provide the actual values of the current/resistance measured.

Our response. We thank the reviewer for raising this point. The current leakage was slightly increased from ~ 390 to $\sim 500 \text{ nA}$ at 1 V , when a PBS solution was put on top of the h-BN layer. However, such a slight increase in leakage current with PBS shows that the h-BN can serve as an effective passivation layer in neural probes.

To clarify this point, we revised the sentence in the revised manuscript, as follows (page 8): “**Only a negligible increase in current leakage was observed, from ~ 390 to $\sim 500 \text{ nA}$ at 1 V** , when a PBS solution was put on top of the h-BN layer.”

Comment 9. On line 158, the authors mention “The average impedance of $\sim 540 \text{ k}\Omega$ is...”. Can you please normalize this per area and compare to the state-of-the-art? Now this must be interesting; would you normalize per $(\pi \cdot r^2)$ with r being the diameter, or $(2 \cdot \pi \cdot r^2)$, considering your electrodes are opened from both sides? Which configuration was the EIS performed? Have you performed EIS before/after resuspension perhaps to see that your impedance decreases when you open up the second part of the electrode?

Our response. We thank the reviewer for raising this point. For a more accurate comparison, we compared the normalized impedance value per area to the work [Nano Lett. 17, 5836 (2017)] that utilized a Pt electrode with a $20 \text{ } \mu\text{m}$ diameter and a single-sided configuration, exhibiting an impedance value of $\sim 1 \text{ M}\Omega$ at 1 kHz . On the other hand, the impedance of the double-sided Pt electrode with a $20 \text{ } \mu\text{m}$ diameter in this work was $\sim 540 \text{ k}\Omega$ at 1 kHz . Normalizing impedance values per area, single-sided and double-sided Pt electrodes with a $20 \text{ } \mu\text{m}$ diameter have similar values of $\sim 314 \times 10^3$ and $\sim 339 \times 10^3 \text{ k}\Omega \cdot \mu\text{m}^2$, respectively.

In addition, the graph below displays the EIS of the double-sided Pt electrodes measured in this work. Compared to previous work [Nano Lett. 17, 5836 (2017)], the double-sided Pt electrode has a lower the impedance at all frequencies than the single-sided Pt electrode.

[Fig. R2] Measurement of EIS.

To respond to the reviewer’s comment, we added the following sentences to the Methods section (‘Evaluating the characteristics of NeuroWeb’) in the revised manuscript: “Normalizing impedance values per area, double-sided Pt electrodes with a 20 μm diameter have a value of $\sim 339 \times 10^3 \text{ k}\Omega \cdot \mu\text{m}^2$, which is comparable to the value in previous work⁴⁹. In addition, in the measured EIS, the double-sided Pt electrode has a lower impedance at all frequencies than the single-sided Pt electrode⁴⁹.”

Comment 10. On line 305, the authors claim “NeuroWeb shows superior performance comparable to implantable probes, while causing no brain tissue damage.” Now this might be an overstatement. How would you safely remove those? Do you leave them inside? Then you must study long-term effect before claiming “no tissue damage”.

Our response. We thank the reviewer for raising this important point. We agree that the term “no brain tissue damage” may not be entirely accurate in describing our NeuroWeb for brain surface recording. However, we would like to emphasize that compared to implantable neural probes, our approach is considerably less invasive.

This argument is supported by the fact that implantable probes require penetration into brain tissue, which can result in tissue damage, inflammation, and a foreign body immune response. In contrast, our NeuroWeb is designed to interface with the cortical surface, thereby reducing the risk of tissue damage and the invasiveness of the treatment.

We believe it is necessary to distinguish between implantable probes and surface recording probes in terms of invasiveness. To more accurately convey this concept and describe the comparative advantage of neural probes, we replaced the term “no brain tissue damage” with “minimal brain tissue damage” in the revised manuscript (page 17).

Comment 11. In Methods (line 671), authors claim that ~30nm thick hBN is etched away in 50W oxygen plasma in 55 seconds, which seems rather unlikely hBN should (a) not be etched away by simple O₂ plasma; definitely not 30nm of it, in MILD plasma for only 55 seconds... At very harsh plasma conditions, the max rate found is ~10-20nm/min [10.1021/acsphotonics.8b00127]. Kindly confirm details of this process step.

Our response. We apologize for confusing the reviewer, regarding the h-BN etching. As the reviewer pointed out, 30 nm-thick h-BN is not completely etched in 50 W oxygen plasma in 55 seconds. In the actual fabrication process, the h-BN was partially etched in this step and fully disappeared when the freestanding structure of NeuroWeb was formed.

To clarify this point, we added a sentence to the Methods section ('Fabrication of NeuroWeb') in the revised manuscript: "The h-BN was partially etched in this step and fully disappeared when the freestanding structure of NeuroWeb was formed."

Comment 12. Finally, I wonder on how do the authors ensure structural integrity of the NeuroWeb while handling the probes? After fabrication and releasing, during EIS/characterization, and during/after implantation? It seems a very technical details but potential paramount to understanding of the probe's potential.

Our response. We thank the reviewer for raising this point. First, after fabricating the NeuroWeb on the wafer, we use optical microscopy to verify that the Gr and Au interconnectors are properly connected. Then, the Ni sacrificial layer is etched to create a freestanding structure of NeuroWeb, and the impedance of a randomly selected NeuroWeb is measured to validate the device characteristics in the same batch. Finally, we measure the impedance after surgery to ensure the structural integrity of NeuroWeb.

To clarify this point, we added a new Methods section "Evaluating the characteristics of NeuroWeb" that includes the above procedure in the revised manuscript.

Response to Reviewer #2.

Comment. In this paper, the authors report an atomically thin, minimally invasive neural probe ("NeuroWeb") composed of hexagonal boron nitride and graphene, which combines the advantages of SEA and iMEA. The authors have realized the open-lattice structure of the NeuroWeb with very thin electrodes only 100 nm thick. The NeuroWeb exhibits high flexibility and strong adhesion, and various experiments have demonstrated that it can be applied to uneven mouse brain surfaces and conformal and tight interfaces. In particular, in in vivo electrophysiological recording experiments, the authors' NeuroWeb detects stable single-unit activity of neurons with a high signal-to-noise ratio. Furthermore, taking advantage of its thinness and high transparency, the authors have successfully combined light stimulation and measurement techniques with electrical measurement techniques. They actually investigated the neural interaction between the somatosensory cortex and the cerebellum using light stimulation and measured the neural signal transmission time between brain regions according to pathways. I acknowledge that the novel thin-film brain electrode NeuroWeb proposed by the authors is an important method to better understand complex brain networks through optical and electrophysiological mapping of the brain. I believe that the novelty of this paper as a brain electrode device and the accuracy of the brain measurements are worthy of publication in Nature Communications, but I would like the authors to answer a few questions before publication to make the paper better. I describe them as follows.

Our response. We thank Reviewer #2 for his/her positive evaluation of the importance of our work. We are happy to have the opportunity to address the reviewer's critical remarks, important questions, and specific suggestions.

Comment 1. A major issue is the susceptibility to noise when the impedance of the measurement probe and its contact impedance to the brain is high. The authors' brain electrodes have great features because they are very thin, but on the other hand, the overall impedance appears to be high. For example, the impedance shown in Extended Data Fig. 4. is several hundred kilo-ohms or more. I believe that the normal impedance for the brain is around a few kilo-ohms. If the impedance is two orders of magnitude higher, the signal-to-noise ratio will be much worse.

Please create a quantitative comparison table with the normal brain probes already reported by other groups to quantitatively demonstrate the usefulness of the NeuroWeb. If NeuroWeb is able to achieve a higher signal-to-noise ratio than other existing probes despite its high impedance, and if NeuroWeb is able to measure cleaner brain signals than conventional probes, please explain the reason in more detail. It is generally known that probes with high impedance are vulnerable to disturbance noise and crosstalk between wires.

Our response. We thank the reviewer for raising this important point regarding the impedance of the probe. However, we respectfully disagree with the reviewer that impedance values in the hundreds of $k\Omega$ have a substantial effect on the signal-to-noise ratio (SNR). It is widely recognized that the impedance of a microelectrode in the range of ~ 0.1 to $2 M\Omega$ has little effect on the data quality and spike sorting in extracellular

recordings [Front. Neurosci. 12, 715 (2018)]. In addition, the impedance values of other probes are also in the range of the hundreds of k Ω (see table below).

We note that NeuroWeb exhibits a relatively high SNR compared to other probes. This result indicates that an impedance of \sim 500 k Ω is not a limitation to a signal measurement with a high SNR. Indeed, NeuroWeb with a thickness of \sim 100 nm detects brain signals more sensitively [Nat. Mater. 9, 511 (2010)] [Nat. Neurosci. 18, 310 (2015)] [Front. Neurosci. 14, 55 (2020)].

To clarify this point, as the reviewer suggested, we added the table for quantitative comparison of SNRs as the **new Supplementary Table 1** to the revised manuscript.

	SNR
This work (NeuroWeb)	9.3
Fu, T. M. et al. [Ref. 38]	8.7
Jun, J. et al. [Ref. 5]	8.1
Kozai, T. et al. [Ref. 9]	8
Shin, H. et al. [Ref. 20]	5.3
Mohanty, A. et al. [Ref. 6]	4.73
Park, S. et al. [Ref. 27]	2.9

[New Supplementary Table 1]

Comment 2. Measuring electrophysiological responses when the brain is stimulated is a very important experiment in the development of brain electrodes. I think that the experimental results shown in Fig. 5 by the authors are excellent. On the other hand, I would like to see the results of Figures 5f and g in more detail, so please add a magnified view. It is important to show directly in the figure what the waveform looks like and after how many seconds it reacts.

Our response. We thank the reviewer for raising this important point. As the reviewer suggested, for better waveform identification, we added magnified views of Figs. 5f and g to the revised manuscript (**new Extended Data Fig. 23**).

[New Extended Data Fig. 23]

Comment 3. For high impedance or thin film electrodes, it is expected that their characteristics will vary greatly with each frequency. Please show the figure with frequency on the horizontal axis and impedance on the vertical axis. If the characteristics differ significantly from frequency to frequency, it will be difficult to measure over a wide frequency range.

Our response. We thank the reviewer for raising this point. As the reviewer suggested, we measured frequency-dependent impedance in the range of 20 Hz to 4000 Hz ($N = 32$ channels): the impedance values were high at low frequencies and low at high frequencies. As shown in Extended Data Fig. 5 in the original manuscript, the capacitance of the Pt electrodes of NeuroWeb results in a high impedance at low frequencies. This is a common feature in other neural probes [Nano Lett. 17, 5836 (2017)] [Nat. Mater. 18, 510 (2019)].

To respond to the reviewer's comment, we added this measured frequency-dependent impedance as the **new Extended Data Fig. 4b** to the revised manuscript.

Comment 4. It is very important for brain research that the electrodes and wiring are transparent and can be linked to the optical system. On the other hand, the figure does not show experimental results that demonstrate quantitative performance regarding transparency. I would like to request the addition of a figure that quantitatively shows transparency as frequency on the horizontal axis and transparency on the vertical axis, as well as a discussion of this issue.

Our response. We thank the reviewer for raising the important point about the transparency of the probe. As the reviewer suggested, we measured the transmittance of the h-BN/Gr/h-BN ribbon over a wide visible wavelength range (480-780 nm). The measurement showed a uniform and high transmittance, with an average transmittance of 96%.

To respond to the reviewer’s comment, we added the measured wavelength-dependent transmittance as the **new Extended Data Fig. 18** to the revised manuscript. In addition, we added a sentence to the main text (page 21): “**We also measured a uniform and high transmittance of NeuroWeb over a wide visible wavelength range (480-780 nm), with an average transmittance of 96% (Extended Data Fig. 18).**”

[New Extended Data Fig. 18]

REVIEWERS' COMMENTS

Reviewer #1 (Remarks to the Author):

The authors addressed all previously raised points, clarifying the manuscript, methods, supporting data and figures. I believe the manuscript may be published as is.

Reviewer #2 (Remarks to the Author):

The authors have carefully addressed the reviewers' suggestions and have revised the paper with important experimental results and discussion. I have no additional questions. The revised paper is, in my opinion, worthy of publication.

Response to Reviewer #1.

Comment. The authors addressed all previously raised points, clarifying the manuscript, methods, supporting data and figures. I believe the manuscript may be published as is.

Our response. We thank Reviewer #1 for his/her positive evaluation of our work and the explicit recommendation for publication.

Response to Reviewer #2.

Comment. The authors have carefully addressed the reviewers' suggestions and have revised the paper with important experimental results and discussion. I have no additional questions. The revised paper is, in my opinion, worthy of publication.

Our response. We thank Reviewer #2 for his/her positive evaluation of our work and the explicit recommendation for publication.